# FINE-GRAINED VERIFIERS: PREFERENCE MODELING AS NEXT-TOKEN PREDICTION IN VISION-LANGUAGE ALIGNMENT

**Chenhang Cui**[1]   **An Zhang**[1][*]  **Yiyang Zhou**[2]   **Zhaorun Chen**[3]   **Gelei Deng**[4]
**Huaxiu Yao**[2]   **Tat-Seng Chua**[1]
[1]National University of Singapore, [2]UNC-Chapel Hill,
[3]University of Chicago, [4]Nanyang Technological University

## ABSTRACT

The recent advancements in large language models (LLMs) and pre-trained vision models have accelerated the development of vision-language large models (VLLMs), enhancing the interaction between visual and linguistic modalities. Despite their notable success across various domains, VLLMs face challenges in modality alignment, which can lead to issues like hallucinations and unsafe content generation. Current alignment techniques often rely on coarse feedback and external datasets, limiting scalability and performance. In this paper, we propose FiSAO (Fine-Grained Self-Alignment Optimization), a novel self-alignment method that utilizes the model's own visual encoder as a fine-grained verifier to improve vision-language alignment without the need for additional data. By leveraging token-level feedback from the vision encoder, FiSAO significantly improves vision-language alignment, even surpassing traditional preference tuning methods that require additional data. Through both theoretical analysis and experimental validation, we demonstrate that FiSAO effectively addresses the misalignment problem in VLLMs, marking the first instance of token-level rewards being applied to such models. Our code is avaliable at `https://github.com/gzcch/FISAO_ICLR`.

## 1 INTRODUCTION

The advent of large language models (LLMs) (Brown et al., 2020; Touvron et al., 2023; Yang et al., 2024) and pre-trained vision models (Radford et al., 2021a; Liu et al., 2023c) has propelled vision-language large models (VLLMs) by advancing connections between visual and linguistic modalities through linear projection (Li et al., 2023b) or q-former (Dai et al., 2023b). These VLLMs have demonstrated notable capabilities across diverse domains such as medical applications (Liu et al., 2023b), autonomous driving (Zhou et al., 2023a), and embodied intelligence (Peng et al., 2023). However, challenges remain in precisely aligning vision and language modalities for integrated inference due to their independent pre-training (Jang et al., 2023; Liu et al., 2024a). This pre-training process often results in incompatible modality-specific representations, hindering the formation of a coherent aligned representation space during joint training (Jang et al., 2023). Misalignment between modalities can lead to safety risks such as biased or inappropriate content generation (Gong et al., 2023; Tu et al., 2023) and hallucinations, where outputs are not grounded in visual input (Wang et al., 2023). These risks are particularly concerning in tasks like visual question answering (Cui et al., 2023; Fan et al., 2024), OCR (Shi et al., 2023), and image captioning (Gunjal et al., 2024), where precise alignment is critical.

To address these misalignment issues, recent works have explored strategies such as instruction tuning (Liu et al., 2023a; Chen et al., 2024b), preference tuning (Yu et al., 2023a), and post-processing methods (Zhou et al., 2023b; Yin et al., 2023). However, most prevalent alignment methods rely heavily on external datasets (Zhou et al., 2024a), models (Yin et al., 2023), or costly human annotations (Yu et al., 2023a). Preference tuning, for example, requires extensive manual labeling, either from human experts (Sun et al., 2023; Yu et al., 2023a) or commercial models (Lee et al., 2023; Li et al., 2023b), which imposes significant costs on building reward datasets and limits scalability.

---

[*]An Zhang is the corresponding author.

Table 1: Feature comparison of different preference tuning approaches.

| Model Name | Reward Model | Additional Data | GPT-Assisted |
|---|---|---|---|
| Vlfeedback (Li et al., 2023d) | × | ✓ | ✓ |
| Human-Preference (Sun et al., 2023) | ✓ | ✓ | × |
| POVID (Zhou et al., 2024a) | × | ✓ | ✓ |
| FiSAO | × | × | × |

Worse still, these alignment methods often rely on coarse feedback, such as sentence-level (Zhou et al., 2024b; Deng et al., 2024) or output-level rewards (Li et al., 2023d), framing the reward modeling task as a simple classification problem that scores outputs as desirable or undesirable. Focusing solely on assigning a numerical score for an entire output fails to leverage VLLMs' token-level generation capabilities, limiting their ability to perform detailed reasoning and precise objective identification.

To mitigate the abovementioned limitations, we propose **F**ine-Gra**i**ned **S**elf-**A**lignment **O**ptimization (**FiSAO**), a method for precisely self-aligning modalities in VLLMs using token-level fine-grained feedback from the vision encoder. Our findings indicate that coarse feedback shows a weak correlation with hallucination detection, while fine-grained reward more effectively differentiates between hallucinated and correct outputs (see Section 3.1). In other words, when using hallucination detection as a proxy for alignment measurement, token-level feedback from the vision encoder offers more informative signals for preference tuning compared to coarse scores. Our theoretical analysis further confirms that this fine-grained feedback improves modality alignment (see Section 3.2). Additionally, FiSAO eliminates the need for external annotations or tools by leveraging its vision encoder as a fine-grained verifier, rewarding each generated token based on its alignment with the visual input. As a result, FiSAO effectively harnesses the model's text generation capabilities and demonstrates superior performance compared to preference tuning methods that rely on additional data. We compare FiSAO with other preference tuning approaches in Table 1.

Our primary contributions can be summarized as follows: We first empirically analyze the differences between coarse and fine-grained rewards in addressing misalignment issues, finding that coarse feedback from pre-trained vision encoders, such as sentence-level rewards, shows a weak correlation with hallucination detection, whereas token-level rewards offer more precise signals for modality alignment. Building on these findings, we propose a novel self-training approach, FiSAO, which leverages token-level feedback from the model's own visual encoder, eliminating the need for additional data or external tools. To the best of our knowledge, FiSAO is the first method to introduce token-level rewards for VLLMs. We further demonstrate FiSAO's effectiveness in mitigating misalignment through both empirical results and theoretical analysis.

## 2 PRELIMINARIES

This section reviews the standard pipeline of preference tuning for VLLMs, as outlined in prior works (Ziegler et al., 2019; Ouyang et al., 2022; Yu et al., 2023a). The process typically consists of three phases: 1) Supervised Fine-Tuning (SFT), 2) Reward Modeling, and 3) Policy Optimization.

**Supervised Fine-Tuning (SFT) Phase.** Preference tuning for VLLMs usually begins by jointly training a pre-trained language model and a pre-trained vision encoder on a high-quality instruction dataset (Li et al., 2023b; Dai et al., 2023b), resulting in a SFT model denoted as $\pi_{\text{SFT}}$.

**Reward Modeling Phase.** Given text $x$ and visual input $v$ as the prompt, the SFT model $\pi_{\text{SFT}}$ is used to generate a pair of responses $(y_1, y_2) \sim \pi_{\text{SFT}}(y|x, v)$. This pair is then evaluated by humans or AI, with one response marked as preferred $y_w$ and the other as less preferred $y_l$, denoted as $y_w \succ y_l|x$. This preference is assumed to follow a latent reward model $r^*(y, x, v)$, which is not directly observable. To model this underlying preference, the Bradley-Terry (BT) model is commonly employed to define the preference distribution $p^*$:

$$p^*(y_w \succ y_l|x) = \frac{\exp(r^*(x, v, y_w))}{\exp(r^*(x, v, y_w)) + \exp(r^*(x, v, y_l))}. \tag{1}$$

Given a static dataset of comparisons $D = \{(x^{(i)}, v^{(i)}, y_w^{(i)}, y_l^{(i)})\}_{i=1}^{N}$ sampled from $p^*$, we can parametrize a reward model $r_\phi(x, v, y)$ and estimate its parameters using maximum likelihood estimation. By formulating the estimation of reward model $r_\phi(x, v, y)$ as a binary classification

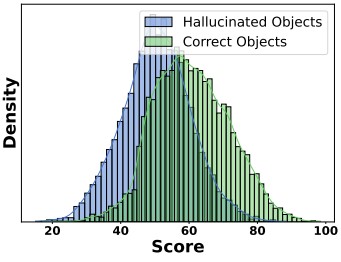 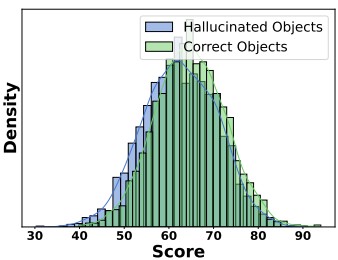

(a) Distributions of Token-Level Reward      (b) Distributions of Sentence-Level Reward

Figure 1: Comparison of token-level (1a) and sentence-level (1b) reward distributions for hallucinated and correct objects in the LLaVA 1.5 model. Further comparisons can be found in Appendix A.2.2.

problem, we define the negative log-likelihood loss $L_R$ as follows:

$$L_R(r_\phi, D) = -\mathbb{E}_{(x,v,y_w,y_l)\sim D}[\log \sigma(r_\phi(x,v,y_w) - r_\phi(x,v,y_l))], \qquad (2)$$

where $\sigma$ denotes the logistic function, and reward model $r_\phi(x,v,y)$ is typically initialized from SFT model $\pi_{\text{SFT}}$, with a linear layer added on top of the final transformer block to produce a scalar output representing the reward prediction (Yu et al., 2023a). Due to the high costs associated with constructing reward model $r_\phi$, such as annotation and training, some preference tuning methods employ external models or tools to directly provide rewards (Hessel et al., 2021).

**Policy Optimization Phase.** The goal of the policy optimization phase is to refine the policy model $\pi_\theta$ using feedback from the reward model $r_\phi$, formulated as:

$$\max_{\pi_\theta} \mathbb{E}_{x,v\sim D, y\sim\pi_\theta(y|x,v)}[r_\phi(x,v,y)] - \beta D_{\text{KL}}[\pi_\theta(y|x,v)||\pi_{\text{ref}}(y|x,v)], \qquad (3)$$

where $\beta$ controls the deviation from the reference policy $\pi_{\text{ref}}$ which is initialized as $\pi_{\text{SFT}}$. This constraint is essential, as it prevents the model from deviating significantly from the original model $\pi_{\text{ref}}$, maintains generation diversity, and prevents mode collapse to high-reward answers (). Due to the discrete nature of language generation, Eqn. 3 is not differentiable. To solve this issue, the standard approach (Ziegler et al., 2019; Ouyang et al., 2022) has been proposed to construct a modified reward function $r(x,v,y) = r_\phi(x,v,y) - \beta(\log\pi_\theta(y|x,v) - \log\pi_{\text{ref}}(y|x,v))$ and then maximize it using Proximal Policy Optimization (PPO) (Schulman et al., 2017).

Although the above preference tuning pipeline enhances models with impressive capabilities (Rafailov et al., 2023), it is considerably more complex than supervised learning, incurring significant computational costs. In light of this, recent alignment methods, such as DPO (Rafailov et al., 2023), have been proposed to streamline the process by conducting preference tuning directly on human-preferred responses without the need for a reward model.

## 3 FISAO

This section first presents empirical findings (Section 3.1), demonstrating that token-level rewards tend to yield improved alignment in Vision-Language Learning Models (VLLMs) compared to sentence-level rewards. A theoretical justification for the effectiveness of FiSAO is then provided in Section 3.2. Following this, Sections 3.3 and 3.4 detail the two-step preference tuning process employed by FiSAO, consisting of reward modeling and policy optimization. The overall framework of FiSAO is illustrated in Figure 3, while Table 1 compares FiSAO with other preference tuning approaches. Unlike other methods, FiSAO eliminates the need for reward model training, additional data, or high-cost human annotators.

### 3.1 EMPIRICAL FINDINGS

Hallucinations in VLLMs occur when these models generate content that is not grounded in the input image (Liu et al., 2024a), such as referencing non-existent objects, often indicating weak alignment between the visual and linguistic modalities (Liu et al., 2024a). To investigate vision-language

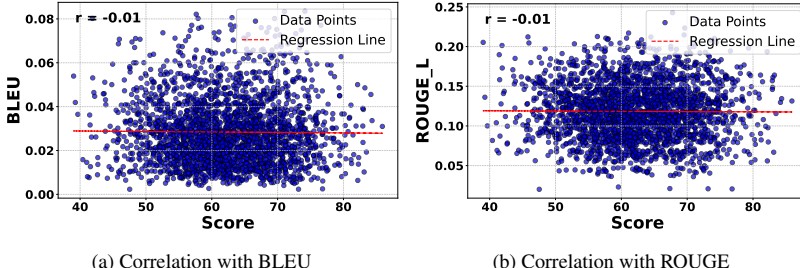

(a) Correlation with BLEU        (b) Correlation with ROUGE

Figure 2: Correlation between the CLIP-based sentence rewards and conventional evaluation metrics: BLEU (2a) and ROUGE (2b). A small Pearson correlation coefficient ($r$) indicates a weak correlation. More comparison is detailed in Appendix A.2.2.

alignment in VLLMs, we examine its relationship to hallucinations. VLLMs commonly extract features using pretrained vision encoders, such as CLIP (Radford et al., 2021a) and Grounding DINO (Liu et al., 2023c). These pretrained vision encoders are trained jointly on vision and language modalities, resulting in a more reliable object recognition (Kuo et al., 2022). Consequently, we propose utilizing the vision encoder of the VLLM as a verifier to investigate two distinct types of reward signals: the sentence-level signal, which is commonly employed in prior research (Hessel et al., 2021; Zhou et al., 2024b), and the token-level signal, which has remained largely unexplored.

To facilitate this investigation, we conducted two experiments: (1) we plot the distribution of scores across the sentence-level and token-level signals for both hallucinated and correctly identified objects, and (2) we examine the relationship between sentence-level rewards and conventional evaluation metrics for VLLMs, such as BLEU and ROUGE. The scores are obtained by calculating the dot product of the text and image embeddings derived from the pretrained vision encoder within the VLLM. We generate captions for 5,000 images randomly sampled from the COCO training dataset and utilize the widely recognized CHAIR hallucination benchmark (Rohrbach et al., 2018) to identify correctly identified and hallucinated objects. We present our observations as follows:

**Token-level rewards differentiate objects better than sentence-level rewards.** Figure 1 presents a comparison of score distributions for hallucinated and correct objects generated by LLaVA-1.5 using two types of rewards: token-level and sentence-level. In the token-level reward distribution (Figure 1a), we observe that hallucinated objects are generally associated with lower scores compared to correct objects. In contrast, in the sentence-level reward distribution (Figure 1b), the two distributions largely overlap, with both hallucinated and correct objects peaking around the same score range (60-70). This indicates that, at the sentence level, the reward signal struggles to distinguish between hallucinated and correct objects.

**Sentence-level rewards show a weak correlation with conventional metrics.** Figure 2 illustrates the relationship between CLIP scores and conventional evaluation metrics BLEU and ROUGE for the generated captions. The scatter plots for BLEU (left) and ROUGE (right) depict the distribution of data points and their corresponding regression lines. From these figures, it is evident that there is a very weak correlation between the scores and both BLEU and ROUGE, with correlation coefficients of $r = -0.01$ for each. Specifically, a high sentence-level score does not necessarily indicate a high-quality sentence. This observation suggests that sentences-level rewards may not be reliable indicators of model performance.

### 3.2 THEORETICAL FRAMEWORK FOR INCORPORATING PRE-TRAINED VISION MODELS' FEEDBACK INTO MODEL TRAINING

In this section, we present a theoretical framework demonstrating how integrating feedback from pre-trained vision models can enhance the performance of VLLMs. Under certain assumptions, we show that utilizing the vision feedback leads to improved quality of model outputs compared to relying solely on supervised fine-tuning.

We consider a VLLM and decompose the input prompt into $x = (v, t) \in \mathbb{R}^{d_v} \times \mathbb{R}^{d_t}$, representing the image and text prompts, respectively. A lthough text data generally consists of discrete tokens, following previous work (Nakada et al., 2023; Chen et al., 2023; Liu et al., 2024d; Zhou et al., 2024b), we model these tokens as continuous random vectors in this section. Specially, we assume

the following data generative model for $v$ and $t$:

$$v = U_v z_v + \xi_v, \quad \text{and} \quad t = U_t z_t + \xi_t, \qquad (4)$$

where $U_v \in \mathbb{O}^{d_v \times r}$ and $U_t \in \mathbb{O}^{d_t \times r}$ are orthonormal matrices representing decoders that transform the latent (low-dimensional) signals $z_v, z_t \in \mathbb{R}^r$ to images and text, respectively. Here, $\xi_v$ and $\xi_t$ are noise vectors, and we assume they follow sub-gaussian distributions with well-conditioned covariance matrices and sub-gaussian norms upper bounded by a universal constant. We consider the infinite data setting, a common simplification to avoid the influence of sample randomness (Kim et al., 2019; Ghorbani et al., 2021; Ye et al., 2023a). According to (Nakada et al., 2023), with an abundance of image-text pairs, the learned visual CLIP embedding $\mathcal{F}_I(v)$ and textual CLIP embedding $\mathcal{F}_T(t)$ converge to $U_v^\top v$ and $U_t^\top t$, respectively. To simplify our analysis without loss of generality, we consider a single score for each response $y$ and define the feedback from pre-trained vision encoders as $R_I(y) = \langle U_v^\top v, U_t^\top y \rangle$. We assume the ground truth $y_{\text{truth}} = V_1^* v + V_2^* t + \epsilon_y$, where $V_1^* \in \mathbb{R}^{d_t \times d_v}$ and $V_2^* \in \mathbb{R}^{d_t \times d_t}$ are weight matrices, and $\epsilon_y$ is a noise term. In our method, we assume that $\pi_{\theta_t}(y \mid x)$ with $\theta_t = (V_1, V_2)$ follows a Gaussian distribution: $\pi_{\theta_t}(y \mid x) \propto \exp\left(-\frac{1}{2\sigma^2}\|y - (V_1 v + V_2 t)\|^2\right)$, where $V_1 \in \mathbb{R}^{d_t \times d_v}$ and $V_2 \in \mathbb{R}^{d_t \times d_t}$ are the weight matrices for the image and text inputs, respectively, and $\sigma > 0$ is the standard deviation.

To better illustrate the contribution of using vision feedback compared to pure supervised fine-tuning (SFT), we consider the supervised fine-tuning score as $R_{\text{sft}}(y) = -\|y - (V_1^* v + V_2^* t)\|^2$. The merged score then becomes

$$R(y) = (1 - \lambda) \cdot R_{\text{sft}}(y) + \lambda \cdot R_I(y), \qquad (5)$$

where $\lambda \in [0, 1]$. As $R(y)$ depends on $\lambda$, we denote the solution $\theta$ by $\theta(\lambda)$. In the special case where $\lambda = 0$, this corresponds to the setting where we do not use feedback from pre-trained vision encoders at all. To assess the quality of the text output $y$, we approach it as a regression problem where there is an associated outcome $z$ linked to the ground-truth text output $y_{\text{truth}}$: $z = \beta^{*\top} y_{\text{truth}}$, with $\beta^* \in \mathbb{R}^{d_t}$. The quality of $y$ is evaluated using the loss function

$$L(y) = \min_{\beta \in \mathbb{R}^{d_t}} \mathbb{E}[(z - \beta^\top y)^2].$$

Note that in this context, a lower value of $L(y)$ indicates better quality of the text output $y$. Consequently, we derive the following theorem.

**Theorem 3.1.** *Suppose that $\pi_{\theta_t}^*(y \mid x)$ lies in the LLM space $\{\pi_\theta(y \mid x) : \theta \in \Theta\}$. Then, there exists some $\lambda > 0$, such that $\mathbb{E}_{\pi_{\theta(\lambda)}(y|x)}[L(y)] < \mathbb{E}_{\pi_{\theta(0)}(y|x)}[L(y)]$.*

The proof can be seen in Appendix A.3.1. Our theoretical analysis implies that integrating feedback from pre-trained vision encoders (where $\lambda > 0$) can enhance VLLMs' performance.

## 3.3 REWARD MODELING FOR FISAO

### 3.3.1 GENERATION FROM THE PERSPECTIVE OF SEQUENTIAL DECISION-MAKING

In this section, we introduce a novel perspective on preference tuning for VLLMs, conceptualizing it as a decision-making process that takes next-token prediction. As discussed in Section 3.1, it is more appropriate to utilize token-level feedback from the fine-grained verifier. Therefore, we consider preference tuning as a decision-making process undertaken by an agent. In this context, after observing the input text and image, a VLLM policy $\pi_\theta$ takes actions by predicting the next token. Here, we consider a standard finite state Markov decision process (MDP) for VLLMs (Puterman, 2014), represented as a tuple $M = (S, A, P, \gamma, R)$. In this context, $S$ is the set of states $s$, representing the current context or history of generated tokens in the VLLM. The set $A$ denotes the actions $a$, which correspond to the possible next tokens that the VLLM can generate. The transition probabilities $P \in \Delta(S)_{S \times A}$ indicate the probability of transitioning from one state to another given an action. The discount factor $\gamma \in (0, 1]$ is typically set to 1 in our case, focusing on the undiscounted scenario. Lastly, $R$ is a bounded reward function $R : S \times A \times S \to \mathbb{R}$, providing feedback or reward for the VLLM $\pi_\theta$ taking action $a$ in state $s$ and transitioning to a new state.

Given an appropriate reward function in $M$, the optimal policy $\pi_M^* \in \Pi$ is the solution to the optimization problem of maximizing the expected discounted total future reward:

$$\max_{\pi \in \Pi} \mathbb{E}_{a_t \sim \pi}\left[\sum_{t=0}^T \gamma^t R(s_t, a_t, s_{t+1})\right]. \qquad (6)$$

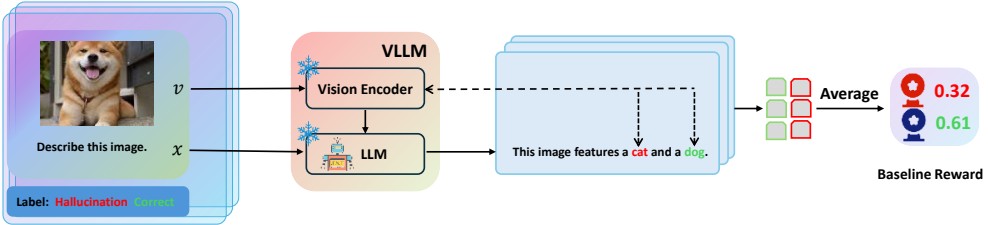

**Step 1  Reward Modeling**

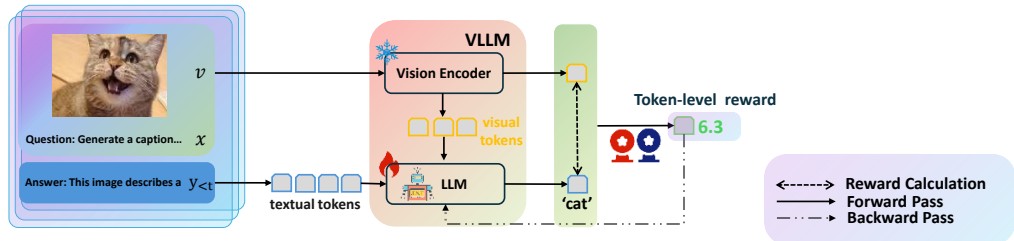

**Step 2 Fine-Grained Preference Policy Optimization**

Figure 3: The overall framework of FiSAO. We employ two steps to achieve self-alignment from fine-grained feedback: (1) calculate the fine-grained reward based on the baseline score obtained from correct and hallucinated tokens. (2) optimize the preference policy using this reward to align the model's responses during training.

This perspective highlights how fine-grained rewards can be applied to enhance and guide VLLMs, enhancing the vision-langauge alignment in VLLMs.

### 3.3.2   ESTIMATION OF BASELINE SCORES FOR GROUND TRUTH AND HALLUCINATED DISTRIBUTIONS

To fairly evaluate the model's performance using feedback from the fine-grained verifier, it is crucial to establish a baseline score. In this section, we estimate the baseline reward for the reward calculation process. Assume that the model generates a set of responses $Y = \{y^1, y^2, \dots, y^s\}$ in response to visual inputs and queries $(x^1, v^1), \dots, (x^s, v^s)$ from the training dataset. Object tokens of these responses can be divided into two subsets: $Y_{gt}$ and $Y_{hal}$. Here, $Y_{gt}$ represents the object tokens that are correctly aligned with the corresponding visual input , determined by the ground truth labels, while $Y_{hal}$ consists of the tokens that are identified as hallucinated or misaligned with the corresponding visual input. For each correct object set $O^i$ and hallucinated object set $\tilde{O}^i$ in $i$-th response, we calculate a score using the dot product between the features of object token and the visual input $v^j$, derived from the fine-grained verifier. Finally, the average scores for correct objects $\mu_{gt}$ and hallucinated objects $\mu_{hal}$ are calculated as follows:

$$\mu_{gt} = \frac{1}{\sum_{i=1}^{s} ||O^i||} \sum_{i=1}^{s} \sum_{o_j \in O^j} S(o_i^j, v^i), \quad \mu_{hal} = \frac{1}{\sum_{i=1}^{s} ||\tilde{O}^i||} \sum_{i=1}^{s} \sum_{o_j \in \tilde{O}^i} S(o_j^i, v^i), \tag{7}$$

where $|| \cdot ||$ denotes cardinality of a set. Eqn. 7 can help define the boundary used to calculate the final reward for fine-grained preference policy optimization.

### 3.3.3   FINE-GRAINED REWARD CALCULATION

In this section, we calculate fine-grained rewards for preference tuning. Formally, let the model's response to a query $x$ with the visual input $v$ from the original dataset be denoted as $\{y_1, y_2, \dots, y_T\}$. To better select tokens suitable for providing feedback, we choose common objects from the existing dataset. First, we construct an entity set using the labels from Detic (Zhou et al., 2022) and COCO (Lin et al., 2015). Then, we expand the original set to $C$ by including similar words and plural forms. Detailed information can be found in the Appendix A.1.1. To better incorporate the feedback from the fine-grained verifier, we calculate the negative and positive reward boundaries based on the baseline

scores of correct and hallucinated responses, as described in Section 3.3.2. We apply the following formula to calculate the fine-grained reward $R = \{R(s_t, a_t, s_{t+1})\}_{t=1}^{T}$:

$$
R(s_t, a_t, s_{t+1}) = \begin{cases} \mathcal{N}(S(y_t, v), (\mu_{\text{hal}} - \lambda)) - \xi D_{\text{KL}}[\pi_{\text{ref}}(x, y_{<t}, v) \| \pi_\theta(x, y_{<t}, v)], \\ \quad \text{if } y_t \in C \& S(y_t, v) < \mu_{\text{hal}} - \lambda \\ \mathcal{N}(S(y_t, v), (\mu_{\text{gt}} + \lambda)) - \xi D_{\text{KL}}[\pi_{\text{ref}}(x, y_{<t}, v) \| \pi_\theta(x, y_{<t}, v)], \\ \quad \text{if } y_t \in C \& S(y_t, v) > \mu_{\text{gt}} + \lambda \\ 0, \qquad \text{otherwise} \end{cases} \tag{8}
$$

where $S(y_t, v)$ is the dot product score of the $y_t$ and $v$ of the pre-trained vision encoder, $\lambda$ is the margin, $\xi$ is a scaling factor for the KL divergence penalty, $\mathcal{N}(\cdot, \cdot)$ is normalization function, $\mu_{\text{gt}}$ and $\mu_{\text{hal}}$ are the average scores of the correct and hallucinated tokens, respectively. More details can be seen in Appendix A.1.5.

## 3.4 FINE-GRAINED PREFERENCE POLICY OPTIMIZATION FOR FiSAO

Following (Ouyang et al., 2022; Yu et al., 2023a), our approach employs a clipped-PPO method to train the model. This method involves cutting the probability ratios to mitigate large updates, ensuring stable and reliable training. Unlike standard PPO, our approach learns from fine-grained feedback at the token level for each state. By incorporating fine-grained preference signals, FiSAO ensures better vision-language alignment in VLLMs. The objective function is defined as:

$$
L(\theta) = \mathbb{E}_{a_t \sim \pi} \left[ \sum_{t=1}^{T} \min \left\{ r_t(\theta), \text{clip}(r_t(\theta), 1 - \epsilon, 1 + \epsilon) \right\} R(s_t, a_t, s_{t+1}) \right], \tag{9}
$$

where $r_t(\theta)$ is the probability ratio, $R_t$ is the advantage estimate and $\epsilon$ is a hyperparameter that determines the clipping range, and clip($\cdot$) is a clipping function that constrains the value of $r_t(\theta)$. The probability ratio $r_t(\theta)$ is calculated as:

$$
r_t(\theta) = \frac{\pi_\theta(y_t | x, y_{<t}, v)}{\pi_{\text{ref}}(y_t | x, y_{<t}, v)}, \tag{10}
$$

where $\pi_{\text{ref}}$ and $\pi_\theta$ are the policies before and after the update, respectively. We show the detailed process of FiSAO in Algorithm 1.

---

**Algorithm 1** FiSAO

---

**Require:** Dataset: $\mathcal{D} = \{(x^i, v^i)\}_{i=1}^{N}$; Reference model: $\pi_{\text{ref}}$; Policy model: $\pi_\theta$; PPO training epochs $e$
**Ensure:** Updated policy model $\pi_\theta$
  1: **for** each $(x, v) \in \mathcal{D}$ **do**
  2:      Generate the response from query and image $\{y_1, y_2, \ldots, y_T\} = \pi_\theta(x, v)$
  3:      **for** each state $y_t$ in $\{y_0, y_1, \ldots, y_T\}$ **do**
  4:          Compute the score $R(s_t, a_t, s_{t+1})$ from Eqn. 8
  5:      **for** each epoch in $e$ **do**
  6:          Calculate probability ratio $r_t(\theta)$ from Eqn. 10
  7:          Update $\pi_\theta$ using Eqn. 9
  8: **return** $\pi_\theta$

---

## 4 EXPERIMENT

In this section, we evaluate FiSAO on the modality alignment of Vision-Language Large Models (VLLMs), showcasing its effectiveness in enhancing models' performance. Our investigation aims to answer the following questions: (1) Does FiSAO enhance the visual understanding capabilities of VLLMs compared to previous approaches? (2) How does the primary component of FiSAO contribute to performance across different benchmarks? (3) Does our method modify the reward distribution of objects in the model's output before and after training?

Table 2: The performance of FiSAO across all benchmarks. **Bold** indicates the best result and underline indicates the second-best result *within each model group* (LLaVA vs. InstructBlip). For CHAIR$_S$ and CHAIR$_I$, smaller is better.

| Method | Comprehensive Benchmark | | | | VQA | | | COCO Benchmark | | |
|---|---|---|---|---|---|---|---|---|---|---|
| | MME$^P$ | MME$^C$ | SEED | MM-Vet | SQA$^I$ | POPE | GQA | Cap_val | CHAIR$_S$ | CHAIR$_I$ |
| LLaVA-1.5 | 1510.7 | 348.2 | 58.6 | 30.5 | 66.8 | 85.9 | 62.0 | 56.6 | 54.3 | 11.3 |
| + Vlfeedback | 1432.7 | 321.8 | 59.3 | 31.2 | 66.2 | 83.7 | **63.2** | 54.8 | 40.3 | 13.2 |
| + Human-Prefer | 1490.6 | 335.0 | 58.1 | 31.1 | 65.8 | 81.5 | 61.3 | 50.4 | 38.7 | 11.3 |
| + POVID | 1452.8 | 325.3 | 60.2 | **31.8** | 68.8 | **86.9** | 61.7 | 57.3 | **35.2** | **8.3** |
| + FiSAO | **1522.6** | **349.0** | **60.6** | 30.5 | **69.3** | 85.7 | 62.0 | **61.2** | 39.9 | 9.9 |
| InstructBlip | 1237.5 | 292.1 | 38.5 | 26.0 | 43.5 | **84.8** | 48.0 | 65.5 | 60.3 | 11.9 |
| + Vlfeedback | 1241.3 | 298.9 | **40.4** | 26.6 | 44.6 | 78.5 | 47.7 | 64.0 | 56.5 | 9.7 |
| + Human-Prefer | 1250.9 | 304.2 | 39.3 | 26.6 | 44.1 | 79.0 | 47.5 | 64.8 | 51.2 | 10.8 |
| + POVID | 1255.1 | 301.8 | 38.3 | 26.3 | 43.4 | 84.6 | **48.3** | 66.5 | 51.5 | 10.5 |
| + FiSAO | **1305.3** | **308.9** | 40.0 | **26.9** | **45.4** | 83.7 | 48.2 | **66.7** | **42.2** | **8.8** |

Table 3: Comparison of FiSAO and other open-sourced state-of-the-art VLLMs.

| Method | MME$^P$ | MME$^C$ | SEED | MMB | MM-Vet | SQA$^I$ | GQA |
|---|---|---|---|---|---|---|---|
| BLIP-2 | 1293.8 | 290.0 | 46.4 | 38.1 | 22.4 | 61.0 | 41.0 |
| InstructBlip | 1237.5 | 292.1 | 38.5 | 36.0 | 26.0 | 43.5 | 48.0 |
| Qwen-VL-Chat | 1487.6 | 360.7 | 58.2 | 60.6 | 47.3 | 68.2 | 57.5 |
| mPLUG-Owl2 | 1450.2 | 313.2 | 57.8 | 64.5 | 36.2 | 68.7 | 56.1 |
| LLaVA-1.5 | 1510.7 | 348.2 | 58.6 | 64.3 | 30.5 | 66.8 | 62.0 |
| FiSAO (LLaVA-1.5) | 1522.6 | 349.0 | 60.6 | 64.8 | 30.5 | 69.3 | 62.0 |

## 4.1 Experimental Setup

**Implementation Details.** We employ LLaVA-1.5 7B (Liu et al., 2024b) and InstructBLIP (Dai et al., 2023b) as the backbone models. During the preference tuning process, we adapt Low-Rank Adaptation (LoRA) (Hu et al., 2021) fine-tuning. We select the first 8k data from the LLaVA-Instruct 150k dataset (Li et al., 2023b). As both InstructBLIP and LLaVA are trained using the LLaVA-Instruct 150k dataset, no additional data is introduced into our model training. Training is conducted over one epoch, with Proximal Policy Optimization (PPO) being applied for four epochs per sample, utilizing four A100 80GB GPUs. Fine-tuning LLaVA-1.5 7B takes approximately six hours, while fine-tuning InstructBLIP 13B requires around ten hours. For more detailed information on training hyperparameters and training data, please refer to Appendix A.1.5.

**Evaluation Benchmarks.** We conduct evaluations on three types of benchmarks: comprehensive benchmarks, general VQA benchmarks and COCO benchmarks. Specifically, these include: (1) Comprehensive benchmarks (MME (Fu et al., 2024), SEEDbench (Li et al., 2023a), MMbench (Liu et al., 2024c), MM-Vet (Yu et al., 2023b)); (2)VQA (ScienceQA (SQA) (Lu et al., 2022), POPE (Li et al., 2023e), GQA (Hudson & Manning, 2019)); (3) Caption benchmark (Li et al., 2024) (Average score of BLEU, ROUGE-L and CIDER), CHAIR (Rohrbach et al., 2019) ). The detailed information is in Appendix A.1.3.

**Baselines.** We compare FiSAO with previous preference tuning approaches, including Silkie (Vlfeedback) (Li et al., 2023d), LLaVA-RLHF (Human-preference) (Sun et al., 2023), and POVID (Zhou et al., 2024a). Furthermore, we compare FiSAO with other state-of-the-art open-source VLLMs, including BLIP-2 (Li et al., 2023c), InstructBLIP (Dai et al., 2023a), Qwen-VL-Chat (Bai et al., 2023), mPLUG-Owl2 (Ye et al., 2023c). More details can be seen in Appendix A.1.4.

## 4.2 Experimental Results on Benckmarks (RQ1)

**Comparison with Other Preference Tuning Approaches.** As shown in Table **??**, our method demonstrates clear advantages over other preference tuning approaches, which often require training reward models or incur high data costs. The superiority of FiSAO lies in its use of fine-grained verifier, which more effectively captures the intrinsic preferences of VLLMs and achieves stronger modality alignment between the pre-trained vison models and LLMs. Additionally, on the LLaVA backbone, FiSAO surpasses existing approaches, delivering an average performance improvement of

8.7%. This underscores FiSAO's effectiveness in leveraging fine-grained token-level rewards to align visual and textual modalities seamlessly.

**Comparison with Other Open-Sourced VLLMs.** Table 3 compares FiSAO with other state-of-the-art VLLMs. Our method, implemented on the LLaVA-1.5 architecture, achieves competitive results across multiple benchmarks, demonstrating its effectiveness in various tasks such as vision question answering and image captioning. This highlights FiSAO's capability in integrating fine-grained token-level rewards to enhance modality alignment in VLLMs.

## 4.3 ANALYSIS (RQ2&RQ3)

**Ablation Study.** Table 4 summarizes the results of the ablation study conducted on FiSAO. Each row represents a different configuration: the presence (✓) or absence (✗) of fine-grained rewards and PPO training. When fine-grained rewards are not used regardless of PPO training, performance metrics are notably lower across all benchmarks compared to configurations where fine-grained rewards are employed. Introducing PPO training alone shows an improvement, but the most significant gains are observed when both fine-grained rewards and PPO training are utilized. This combination achieves the highest scores, demonstrating the effectiveness of integrating both strategies in enhancing model performance and alignment across various evaluation tasks. These findings underscore the importance of fine-grained token-level rewards in optimizing VLLMs such as FiSAO for multimodal tasks.

**How does Reward Margin Effect Model's Performance?** We present how different reward margins impact the model's performance across various benchmarks in Table 5. The table highlights how varying the reward margin $\lambda$ affects the performance of LLaVA-1.5 + FiSAO across multiple benchmarks. The results indicate notable variations in performance metrics based on the choice of reward margin. Specifically, when the margin is either too small or too large, a decline is observed in metrics such as CHAIR$_I$, suggesting diminishing returns with extreme reward margins. Although overall performance remains relatively stable, these findings underscore the importance of optimizing the reward margin to balance precision and generalization in FiSAO for enhancing the performance of VLLMs.

**How does FiSAO Alter the Reward Distribution of Objects in the Model's Output before and after Training?** To better demonstrate how our method enhances vision-language alignment and ensures the generation of high-scoring objects, we visualize the reward distribution of generated objects on the CHAIR benchmark, as depicted in Figure 4. The figure illustrates that VLLMs tend to generate objects with lower scores before training. This result indicates that the reward distribution before training is more dispersed and misaligned with the preferences of the visual encoder. After applying our method, the reward distribution shifts to the right, reflecting improved vision-language alignment. This shift signifies that fine-grained feedback leads to enhanced overall performance in VLLMs.

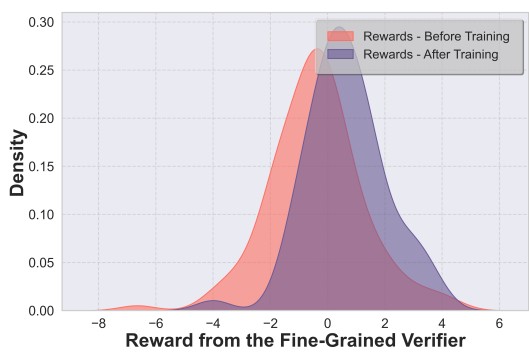

Figure 4: Comparison of reward distributions for generated objects on LLaVA-1.5 before and after Training.

**Case Study on Sentence-Level Reward and Token-Level Reward.** In this section, we conduct a case study where two sentences from an image are selected for evaluation using both token-level and sentence-level scoring. From Figure 5, we can observe that the sentence-level score is not sensitive to hallucinatory sentences, as it assigns similar scores to both sentences. In contrast, token-level scoring more effectively identifies hallucinatory objects.

## 5 RELATED WORK

Recent advancements in large language models (Brown et al., 2020; Liu et al., 2022; Touvron et al., 2023) and pre-trained vision models (Radford et al., 2021a) have enabled the creation of Vision-Large

The image features a **woman** and a young girl walking down a street together. Both of them are wearing **umbrellas**, with the woman holding one umbrella and the girl holding another umbrella. The woman appears to be guiding the girl as they walk.
Hallucination: ✗
Sentence-Level Score: 0.547
Token-Level Score: **woman**: 0.576 **umbrellas**: 0.718

-------------------------------------------------------------------

There are several other **people** in the scene, some of them standing or walking nearby. In total, there are three other individuals in the image, some of them closer to the woman and the girl. Additionally, there is a **handbag** visible in the scene, likely belonging to one of the people present.
Hallucination: √
Sentence-Level Score: 0.559
Token-Level Score: **handbag**: 0.374 **people**: 0.636

Figure 5: Case study on sentence-level reward and token-level reward.

Table 4: Ablation study results. Each row illustrates a different configuration, indicating the presence (✓) or absence (✗) of fine-grained rewards and PPO training.

| | | Comprehensive Benchmark | | | | | VQA | | | COCO-cap Benchmark | | |
|---|---|---|---|---|---|---|---|---|---|---|---|---|
| Fine-grained | PPO | $MME^P$ | $MME^C$ | SEED | MMB | MM-Vet | $SQA^I$ | POPE | GQA | Cap_val | $CHAIR_S$ | $CHAIR_I$ |
| ✗ | ✗ | 1431.9 | 340.0 | 59.6 | 64.0 | 30.6 | 67.7 | 85.7 | 61.4 | 54.5 | 54.0 | 11.0 |
| ✗ | ✓ | 1509.3 | 350.4 | 59.5 | 64.1 | 30.5 | 67.5 | 85.9 | 60.9 | 56.6 | 55.3 | 11.4 |
| ✓ | ✓ | 1522.6 | 349.0 | 60.6 | 64.8 | 30.7 | 69.3 | 85.7 | 62.0 | 61.2 | 39.9 | 9.9 |

Table 5: Performance of FiSAO with varying margins

| | Comprehensive Benchmark | | | | | VQA | | | COCO-cap Benchmark | | |
|---|---|---|---|---|---|---|---|---|---|---|---|
| $\lambda$ | $MME^P$ | $MME^C$ | SEED | MMB | MM-Vet | $SQA^I$ | POPE | GQA | Cap_val | $CHAIR_S$ | $CHAIR_I$ |
| 5 | 1509.3 | 350.4 | 60.4 | 64.1 | 30.6 | 67.5 | 84.4 | 61.7 | 57.1 | 53.3 | 10.8 |
| 10 | 1522.6 | 349.0 | 60.6 | 64.8 | 30.7 | 69.3 | 85.7 | 62.0 | 61.2 | 39.9 | 9.9 |
| 20 | 1501.4 | 348.6 | 59.2 | 64.5 | 31.0 | 67.9 | 85.1 | 61.6 | 59.7 | 56.5 | 13.6 |

Language Models (VLLMs), which effectively integrate language and vision capabilities. These models have significantly improved automation in various fields, including medical applications (Liu et al., 2023b), recommendation (Sheng et al., 2024), autonomous driving (Zhou et al., 2023a), agent-based evaluation (Zheng et al., 2024),and embodied agents (Peng et al., 2023). The typical architecture of VLLMs involves aligning the embedding spaces of both modalities using techniques such as Qformer or fully connected layers (Zhu et al., 2023; Ye et al., 2023b; Li et al., 2023b). However, VLLMs face challenges in precise alignment due to independent pre-training of language and vision models, leading to safety concerns (Gong et al., 2023; Tu et al., 2023), hallucinations (Wang et al., 2023), and reasoning deficiencies (Ghosh et al., 2024). Traditional vision-language models (VLMs) have focused on image-text alignment through methods like co-attention frameworks (Lu et al., 2019), anchor points (Li et al., 2020), and contrastive learning (Radford et al., 2021b). Recently, alignment strategies can be classified into alignment from training data, which leverages high-quality datasets for supervised fine-tuning (SFT), and alignment from feedback, which involves fine-tuning based on human or AI feedback (Sun et al., 2023; Yu et al., 2023a; Zhou et al., 2024a; Li et al., 2023d; Zhao et al., 2023). Feedback-based methods often use Proximal Policy Optimization (PPO) (Sun et al., 2023) and Direct Preference Optimization (DPO) (Zhao et al., 2023; Li et al., 2023d; Chen et al., 2024a). Despite their potential, these methods face challenges such as high costs in dataset construction and need for external tools. Additionally, some approaches use sentence-level rewards, which do not fully leverage the text-generation capabilities that large language models (LLMs) are fundamentally designed for. By concentrating on assigning a numerical score to the entire instance, these methods overlook the VLLMs' inherent capability to generate responses, including detailed reasoning steps. The detailed version is shown in Appendix A.4.

## 6 CONCLUSION

In this study, we addressed the alignment issues prevalent in Vision-Language Large Models (VLLMs) by investigating the integration of pre-trained vision encoders with large language models. Through comprehensive analysis, we introduced a novel self-training method using fine-grained Proximal Policy Optimization (PPO) that does not rely on additional data. This method leverages the model's visual encoder as a reward model to enhance alignment at the token level, demonstrating superior performance compared to existing preference tuning approaches.

## ACKNOWLEDGEMENT

This research is supported by the NExT Research Centre and Special Funding for Students' Overseas Research Internships of University of Electronic Science and Technology of China (UESTC). We sincerely appreciate the reviewers and the AC for their valuable suggestions throughout the review process.

## ETHICS STATEMENT

This paper aims to enhance vision-language alignment for Vision-Language Large Models (VLLMs) and do not obey the ICLR code of ethics.

## REPRODUCIBILITY STATEMENT

All the results in this work are reproducible. We provide detailed settings for our experiments in Table 7.

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

# A APPENDIX

## A.1 EXPERIMENTAL SETTINGS

### A.1.1 DETAILS OF ENTITY SET

First, we construct an entity set using the labels from Detic (Zhou et al., 2022) and COCO (Lin et al., 2015). We present the case of these datasets' labels in Table 6 Then, we expand the original set to $C$ by including similar words and plural forms using the `inflect` library and the `wordnet` module from the `nltk` library. The expanded set $C$ contains 5678 words compared to the original set, which contains 1204 words. The `inflect` library is used to generate plural and singular forms of the original labels, while the `wordnet` module from `nltk` is employed to find synonyms. This method allows us to create a comprehensive entity set by considering various linguistic forms, thus enhancing the robustness of our dataset.

| Original Word | Expanded Words |
|---|---|
| apple | apples |
| handbag | bag, handbags, pocketbook, purse |
| suitcase | grip, suitcases |
| bagel | bagels |
| boat | boats, sauceboat, boat |
| bob | dock, cork, bobs |
| bread | breads, lucre, lolly, staff of life |
| cat | purge, chuck, cats |
| chair | moderate, chairs, chairperson, lead, chairman |
| duck | ducks, duck, dip, douse |
| jar | jars, clash, shock |
| person | someone, person, individual, somebody, people, soul |
| shirt | shirts |
| taco | tacos, greaser, wetback, taco |
| wheel | cycle, wheels, roll |

Table 6: Cases of original Words and their expanded forms.

### A.1.2 OVERVIEW OF THE BACKBONE MODELS

**LLaVA-1.5** is a multimodal model designed for general-purpose visual and language understanding. It integrates a vision encoder with the Vicuna language model, making it capable of processing images and generating text-based responses. The model is an open-source chatbot that has been fine-tuned on multimodal instruction-following data generated by GPT. It is built upon the transformer architecture, specifically leveraging the LLaMA/Vicuna foundation models.

**InstructBLIP** is a sophisticated vision-language model designed to follow detailed instructions. It is built upon the BLIP-2 architecture, incorporating a vision encoder, a language model, and a Query Transformer (Q-Former) that bridges the two components. The Q-Former module is specifically enhanced to handle instruction text tokens, allowing it to extract task-relevant features from images effectively.

### A.1.3 DETAILS OF EVALUATION BENCHMARK

- MME (Fu et al., 2024) is a comprehensive benchmark for evaluating the performance of LVLMs in multimodal tasks. It measures models' capabilities across two key areas: perception and cognition, using 14 specially designed subtasks that test interpretative and analytical skills.

- SEED-Bench (Li et al., 2023a) focuses on evaluating the generative comprehension abilities of LVLMs. It includes a dataset of 19K multiple-choice questions with detailed human annotations,

spanning 12 evaluation dimensions that cover both spatial and temporal understanding in image and video modalities.

- MMBench (Liu et al., 2024c) employs a dual approach: it provides an extensive dataset that broadens the range and variety of evaluation questions, and introduces the innovative CircularEval strategy, which uses ChatGPT to convert free-form predictions into structured choices.

- MM-Vet (Yu et al., 2023b) is a benchmark created to evaluate the diverse competencies of LVLMs. It organizes complex multimodal tasks into 16 unique integrations based on six core vision-language capabilities, offering a detailed analysis of model performance across various question types and answer styles.

- ScienceQA (Lu et al., 2022) is a multimodal benchmark aimed at assessing and diagnosing AI systems' multi-hop reasoning and interpretability in the science domain. It includes a dataset of around 21K multiple-choice questions across various scientific topics, complete with detailed answer annotations, related lectures, and explanations.

- GQA (Hudson & Manning, 2019) is a dataset designed for advanced visual reasoning in real-world scenarios, using scene graph-based structures to generate 22 million diverse, semantically-programmed questions. It features a novel set of evaluation metrics focused on consistency, grounding, and plausibility, setting a high standard for vision-language task assessment.

- POPE (Li et al., 2023e) is an evaluation method for examining object hallucination in LVLMs. It transforms the evaluation into a binary classification task, asking LVLMs simple Yes-or-No questions to identify hallucinated objects. POPE employs various object sampling strategies to reveal model tendencies towards hallucination.

- The COCO-caption benchmark assesses image captioning models using BLEU, ROUGE, and CIDEr scores, providing a comprehensive measure of caption quality. We calculate the average of these scores and multiply by 100 to obtain the final score. This benchmark utilizes the COCO dataset, emphasizing the accuracy and relevance of generated captions. Detailed evaluation methodology and task specifics can be found in the `lmms_eval` repository, specifically under the `tasks/coco2017_cap_val` directory.[1]

- CHAIR (Rohrbach et al., 2019) is a well-known tool for evaluating object hallucination in image captioning tasks. It includes two variants: CHAIRI and CHAIRS, which assess object hallucination at the instance and sentence levels, respectively. Specifically, we randomly sampled 500 images from the COCO (Lin et al., 2015) validation set and evaluated object hallucination using the CHAIR metric.

### A.1.4 DETAILS OF BASELINES

- Silkie (Vlfeed- back) (Li et al., 2023d) focuses on improving large vision language models (LVLMs) by using preference distillation. The authors created a vision-language feedback (VLFeedback) dataset, consisting of multi-modal instructions and responses generated by 12 different LVLMs. The model pool includes prominent models like GPT-4V and LLaVA-series. By applying direct preference optimization (DPO) on this dataset, they developed the Silkie model, which shows significant improvements in perception and cognition capabilities.

- LLaVA-RLHF (Human-preference) (Sun et al., 2023) explores the integration of reinforcement learning with human feedback (RLHF) to enhance vision-language models. The LLaVA series, built on Vicuna models and fine-tuned with GPT-4 generated multi-modal data, is further improved by aligning visual faithfulness and human preferences. This approach aims to ensure that the generated responses are more aligned with human expectations and the visual content they describe, providing a more reliable and contextually accurate output

---

[1]https://github.com/EvolvingLMMs-Lab/lmms-eval/tree/main/lmms_eval/tasks

| Backbone Model | LLaVA-1.5 | InstructBLIP |
|---|---|---|
| Parameter | 7B | 13B |
| Reward Model | CLIP-ViT-L-334 | CLIP-ViT-L |
| Dataset | LLaVA-Instruct | LLaVA-Instruct |
| Fine-Tuning Method | LoRA | LoRA |
| Number of Epochs | 1 | 1 |
| PPO Training Epochs | 4 | 4 |
| GPUs Used | 4 A100 80GB GPUs | 4 A100 80GB GPUs |
| Training Time | ~6 hours | ~10 hours |
| LoRA r | 128 | 128 |
| LoRA Alpha | 256 | 256 |
| Learning Rate | 5e-7 | 4e-6 |
| LoRA Parameter | all linear | all linear |
| $\xi$ | 0.2 | 0.2 |
| $\lambda$ | 10 | 10 |

Table 7: Training parameters for LLaVA-1.5 7B and InstructBLIP 13B models.

- POVID (Zhou et al., 2024a) is a framework for generating non-preferred responses in Vision-Language Large Models (VLLMs) aimed at preference optimization. The framework employs two strategies: hallucination text responses and noisy image responses at token and instance levels. This approach helps in understanding and optimizing VLLMs by intentionally producing outputs that are less preferred, thus identifying areas for improvement in model performance and user interaction.

### A.1.5 HYPERPARAMETER DETAILS

In this section, we show the detailed information on training hyperparameters and training data in Table 7. Specifically, for the normalized function $\mathcal{N}(\cdot, \cdot)$, we calculate the score for correct objects as $\frac{S(y_t, v) - (\mu_{gt} + \lambda)}{S_{\max} - (\mu_{gt} + \lambda)}$, and for hallucinated objects as $\frac{S(y_t, v) - (\mu_{hal} - \lambda)}{(\mu_{gt} - \lambda) - S_{\min}}$. $S_{\min}$ and $S_{\max}$ represent the minimum and maximum possible scores, respectively. In this way, we constrain the reward within the range of $-1$ to $1$.

### A.2 ADDITIONAL ANALYSIS

#### A.2.1 DETAILED ANALYSIS ON COCO-CAPTION BENCHMARK

Table 8 provides a comprehensive comparison of various methods evaluated on COCO-caption benchmark. Our method, denoted as FiSAO, demonstrates significant improvements across multiple metrics, highlighting its efficacy in enhancing caption generation quality. On the LLaVA backbone, FiSAOconsistently outperforms the baseline and other preference-tuning methods across all BLEU metrics, as well as METEOR, ROUGE L, and CIDEr scores. These results underscore the robustness of FiSAOin capturing nuanced textual and visual features, achieving superior alignment and coherence in the generated captions. Similarly, for the InstructBLIP backbone, FiSAOmaintains a competitive edge, achieving high scores across the evaluation metrics and outperforming other preference-tuning approaches. The improvements observed with FiSAOhighlight its effectiveness in leveraging fine-grained token-level rewards to enhance the alignment between visual and textual modalities.

#### A.2.2 ADDITIONAL ANALYSIS ON SENTENCE-LEVEL REWARD

We present the sentence-level rewards of the generated captions on InstructBLIP in Figure 6. We can observe the low distinction between correct and hallucinated captions. We also show comparison of Fine-Grained and sentence-level reward distribution in Figure 7 and Figure 8, where the sentence-level reward shows no explicit correlation with traditional evaluation scores. This comparison highlights that the Fine-Grained reward distribution tends to be more useful, offering a detailed view of the

Table 8: Evaluation results on COCO-caption benchmark.

| Method | Bleu 1 | Bleu 2 | Bleu 3 | Bleu 4 | METEOR | ROUGE L | CIDEr |
|---|---|---|---|---|---|---|---|
| LLaVA | 0.7312 | 0.5641 | 0.4150 | 0.2976 | 0.2929 | 0.5559 | 1.1038 |
| + Vlfeedback | 0.7149 | 0.5487 | 0.3734 | 0.2788 | 0.2835 | 0.5398 | 1.0969 |
| + Human-Prefer | 0.6741 | 0.5047 | 0.3613 | 0.2519 | 0.2864 | 0.5329 | 0.9142 |
| + POVID | 0.7360 | 0.5680 | 0.4197 | 0.3030 | 0.2954 | 0.5601 | 1.1305 |
| + FiSAO | 0.7925 | 0.6259 | 0.4681 | 0.3407 | 0.2811 | 0.5774 | 1.1970 |
| InstructBLIP | 0.8220 | 0.6682 | 0.5199 | 0.3973 | 0.2982 | 0.5984 | 1.3498 |
| + Vlfeedback | 0.7919 | 0.6346 | 0.4886 | 0.3689 | 0.3000 | 0.5874 | 1.3055 |
| + Human-Prefer | 0.8034 | 0.6431 | 0.5068 | 0.3759 | 0.3104 | 0.6012 | 1.2902 |
| + POVID | 0.8204 | 0.6671 | 0.5198 | 0.3977 | 0.3009 | 0.6002 | 1.3619 |
| + FiSAO | 0.8239 | 0.6707 | 0.5231 | 0.4008 | 0.2985 | 0.5994 | 1.3526 |

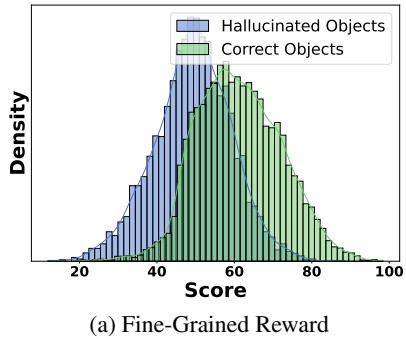

(a) Fine-Grained Reward

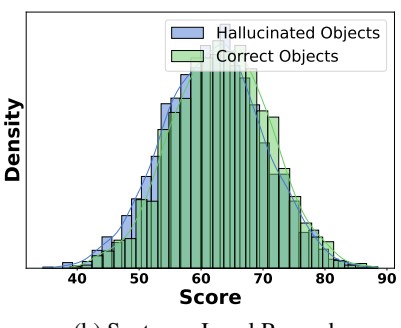

(b) Sentence-Level Reward

Figure 6: Comparison of fine-grained and sentence-level reward distributions in InstructBLIP.

model's performance. These analyses further demonstrate that using Fine-Grained rewards is more effective than sentence-level rewards.

### A.2.3 ADDITIONAL ANALYSIS ON REWARD DISTRIBUTION OF OBJECTS

To further illustrate how our method enhances the alignment between visual encoders and VLLMs, we present the reward distribution of hallucinated objects in Figure Figure 9. The figure shows that, before training, the reward distribution for hallucinated objects in both LLaVA and InstructBLIP is more scattered and less aligned with the visual encoder's preferences. After applying our method, the reward distribution shifts to the right, indicating improved alignment and consistency with the visual encoder. This shift demonstrates that the model's rewards now more accurately reflect the visual encoder's evaluations, thereby enhancing the overall performance of vision-language alignment.

### A.2.4 CASE STUDIES

In this section, we present detailed case studies comparing the outputs of our model with LLaVA 1.5. The case studies highlight the strengths of FiSAOin generating detailed image descriptions. As shown in Figure 10, FiSAOfocuses on providing a comprehensive overview, including contextual details such as the environment and the placement of objects (e.g., handbag, table settings). This approach ensures that the description covers all relevant aspects of the scene. LLaVA 1.5 includes specific interactions and objects that enhance the vividness of the scene. However, it sometimes generates objects that are not actually present in the images.

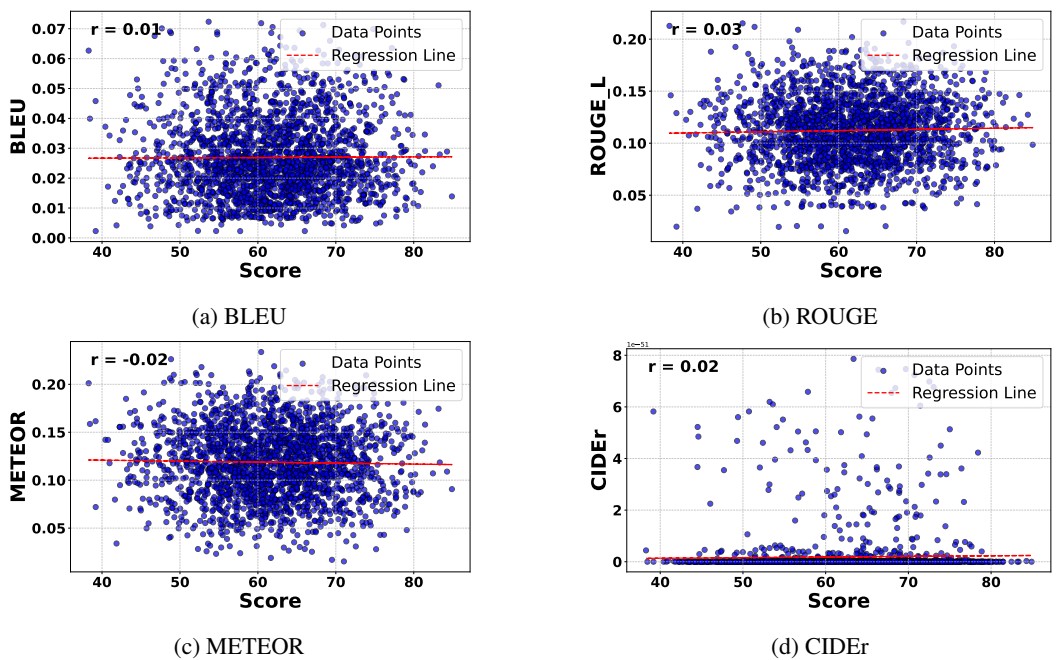

Figure 7: Correlation between sentence reward and conventional evaluation metrics on InstructBLIP.

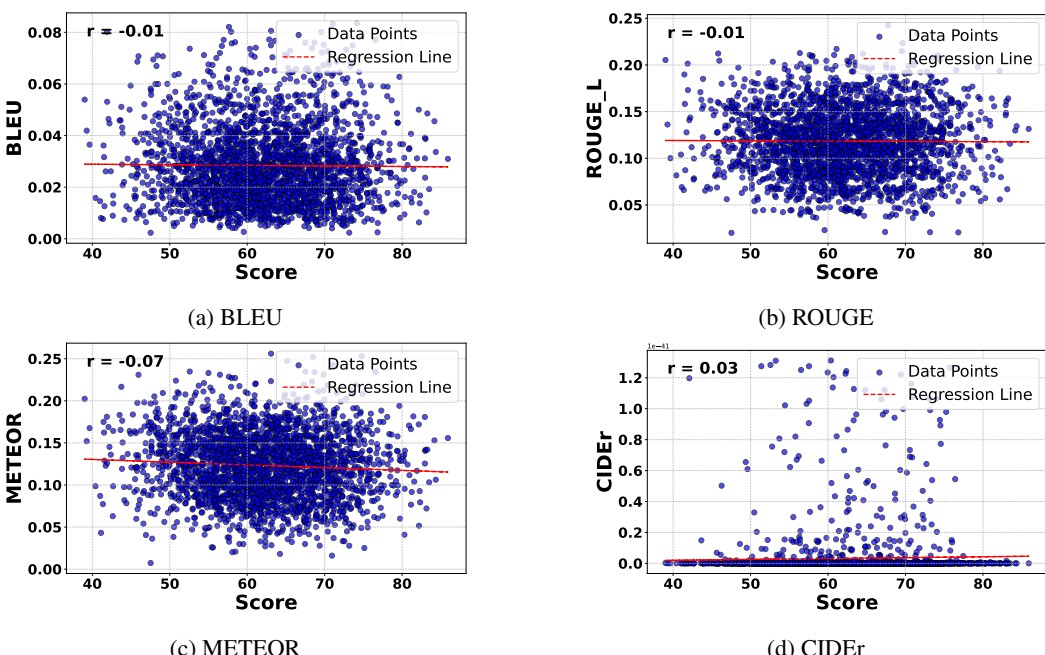

Figure 8: Correlation between sentence reward and conventional evaluation metrics on LLaVA

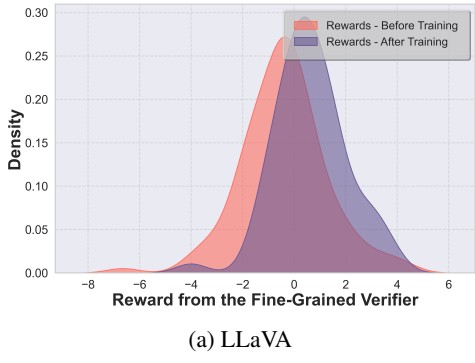 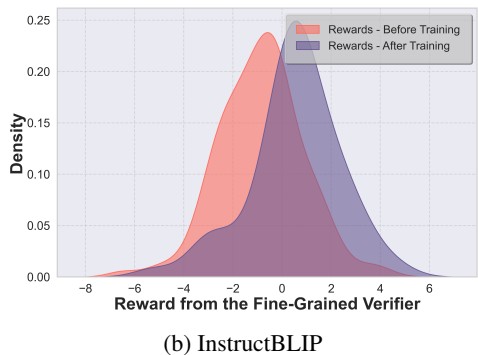

(a) LLaVA       (b) InstructBLIP

Figure 9: Reward distribution comparison before and after training.

Prompt: Describe this image in detail.

       Ours         LLaVA 1.5

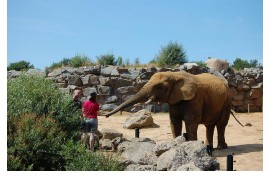

The image features a group of people standing near a large elephant. There are at least five people in the scene, with some standing closer to the elephant and others further away. The elephant is positioned towards the right side of the image, taking up a significant portion of the frame.

In addition to the people and the elephant, there is a handbag located near the left side of the image, possibly belonging to one of the people in the group. The scene appears to be set in a dirt area, which is a common environment for elephants.

The image features a group of people standing near a large elephant in a dirt area. The elephant is positioned in the center of the scene, with its trunk extended towards the people. There are at least five people in the scene, with some standing closer to the elephant and others further away. The people are interacting with the elephant, with one person feeding it a **carrot**. The **carrot** is visible in the elephant's mouth, and the person is holding it in their hand. The other people are observing the interaction, likely enjoying the unique experience of being so close to such a majestic creature.

Prompt: Describe this image in detail.

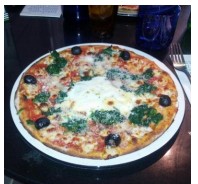

The image features a delicious pizza sitting on a dining table. The pizza is topped with a variety of ingredients, including cheese, olives, and spinach. There are several olives scattered across the pizza, with some located near the center and others towards the edges. The spinach is also evenly distributed across the pizza, adding a vibrant green color to the dish.

In addition to the pizza, there are two cups on the table, one near the top left corner and the other near the top right corner. A fork is also present on the table, positioned towards the right side. The table occupies most of the image, with the pizza being the main focus.

The image features a delicious pizza with a variety of toppings, including olives, cheese, and spinach, placed on a dining table. The pizza is cut into slices and is ready to be eaten. The table is set with a fork and a **knife**, indicating that the pizza is meant to be enjoyed with utensils.

In addition to the pizza, there are two cups on the table, one near the top left corner and the other near the top right corner. A **bowl** is also present on the table, located towards the top right corner. The dining table occupies most of the image, with the pizza taking up a significant portion of the table's surface.

Figure 10: Case studies on LLaVA 1.5.

## A.3   WHY DOES FEEDBACK FROM PRETRAINED VISION ENCODERS CONTRIBUTE TO THE MODEL'S PERFORMANCE - THEORETICAL ANALYSIS

### A.3.1   PROOF OF THEOREM 3.1

We begin by considering the distribution of the generated response $y_p$ given by $\pi_{\theta_t}^*(y \mid x)$. Since $y_p = \arg\max_y R(y)$, this distribution is a point mass. The global minimizer will converge to $\pi_{\theta_t}^*(y \mid x)$.

By our method, we have

$$y_p = \arg\max_y \left[ (1 - \lambda) \left( -\|y - (V_1^* v + V_2^* t)\|^2 \right) + \lambda \langle U_v^\top v, U_t^\top y \rangle \right]. \tag{11}$$

Simplifying, we rewrite the optimization problem as

$$y_p = \arg\min_y \left[ \|y - (V_1^* v + V_2^* t)\|^2 - \gamma \langle U_v^\top v, U_t^\top y \rangle \right],$$

where $\gamma = \frac{\lambda}{1-\lambda}$. Taking the derivative with respect to $y$ and setting it to zero yields

$$2\left(y - (V_1^* v + V_2^* t)\right) - \gamma U_t U_v^\top v = 0.$$

Solving for $y_p$, we obtain

$$y_p = (V_1^* v + V_2^* t) + \frac{\gamma}{2} U_t U_v^\top v.$$

This shows that integrating vision feedback effectively increases the weight on the visual input.

Next, we consider the loss function

$$L(y) = \min_{\beta \in \mathbb{R}^{d_t}} \mathbb{E}\left[\left(z - \beta^\top y\right)^2\right], \tag{12}$$

where $z = \beta^{*\top} y_{\text{truth}}$ and $y_{\text{truth}} = V_1^* v + V_2^* t + \epsilon_y$.

Substituting the expressions for $y_p$ and $y_{\text{truth}}$, we have

$$L(y_p) = \min_\beta \mathbb{E}\left[\left(\beta^{*\top} y_{\text{truth}} - \beta^\top y_p\right)^2\right]. \tag{13}$$

Expanding, we get

$$L(y_p) = \min_\beta \mathbb{E}\left[\left((\beta^{*\top} - \beta^\top)(V_1^* v + V_2^* t) - \beta^\top \left(\frac{\gamma}{2} U_t U_v^\top v\right) + \beta^{*\top} \epsilon_y\right)^2\right]. \tag{14}$$

We introduce an assumption that $\epsilon_y$ contains a component that can be estimated via vision feedback. Suppose

$$\epsilon_y = \kappa U_t U_v^\top v + \tilde{\epsilon}, \tag{15}$$

where $\tilde{\epsilon}$ is noise independent of $v$, and $\kappa$ is a scalar.

Therefore,

$$y_{\text{truth}} = V_1^* v + V_2^* t + \kappa U_t U_v^\top v + \tilde{\epsilon}. \tag{16}$$

Now, since

$$y_p^{(\lambda)} = V_1^* v + V_2^* t + \frac{\gamma}{2} U_t U_v^\top v, \tag{17}$$

the vision feedback term helps to estimate part of $\epsilon_y$.

We define the mean squared error:

$$\text{MSE}_\lambda = \mathbb{E}\left[\left\|y_p^{(\lambda)} - y_{\text{truth}}\right\|^2\right]. \tag{18}$$

Substituting,

$$\text{MSE}_\lambda = \mathbb{E}\left[\left\|\left(\frac{\gamma}{2} - \kappa\right) U_t U_v^\top v - \tilde{\epsilon}\right\|^2\right]. \tag{19}$$

For $\lambda = 0$,

$$\text{MSE}_0 = \mathbb{E}\left[\left\|-\kappa U_t U_v^\top v - \tilde{\epsilon}\right\|^2\right]. \tag{20}$$

The difference is

$$\Delta\text{MSE} = \text{MSE}_\lambda - \text{MSE}_0 = \left[\left(\frac{\gamma}{2} - \kappa\right)^2 - \kappa^2\right] \mathbb{E}\left[\left\|U_t U_v^\top v\right\|^2\right]. \tag{21}$$

Setting $\gamma = 2\kappa$ (which implies $\lambda = \frac{2\kappa}{2\kappa+1} > 0$), we have

$$\Delta\text{MSE} = -\kappa^2 \mathbb{E}\left[\left\|U_t U_v^\top v\right\|^2\right] < 0. \tag{22}$$

Thus, there exists $\lambda > 0$ such that

$$\mathbb{E}_{\pi_{\theta(\lambda)}(y|x)}[L(y)] < \mathbb{E}_{\pi_{\theta(0)}(y|x)}[L(y)]. \tag{23}$$

This proves the theorem.

By selecting a suitable $\lambda > 0$, we have demonstrated that integrating vision feedback can reduce the expected loss. Therefore, incorporating vision feedback helps the model to predict the output more accurately, which proves Theorem 3.1.

## A.4 RELATED WORK

### A.4.1 VISION-LARGE LANGUAGE MODEL

Recently, the development of large language models (Brown et al., 2020; Touvron et al., 2023)and pre-trained vision models (Radford et al., 2021a), has paved the way for Vision-Large Language Model(VLLMs). These advanced models, which can comprehend both text and images, have greatly enhanced our capacity to automate complex tasks accross various areas such as medical application (Liu et al., 2023b), autonomous driving (Zhou et al., 2023a) and embodied agent (Peng et al., 2023). The fundamental architecture of VLLMs typically integrates both language and vision models. This integration involves aligning the embedding spaces of both modalities using Qformer or a simple fully connected layer (Zhu et al., 2023; Ye et al., 2023b; Li et al., 2023b). However, Vision-Language Large Models (VLLMs) still face the problem of misalignment, as both models are typically pre-trained independently before being aligned through vision-language joint training. This misalignment can lead to several issues, such as safety concerns, where the model may produce inappropriate or biased content (Gong et al., 2023; Tu et al., 2023), hallucinations in VLLMs, where the model generates information not grounded in the images, thus deviating from observable reality (Wang et al., 2023), and deficiencies in logical reasoning (Ghosh et al., 2024), where the model fails to coherently integrate visual and textual information, resulting in inaccurate outputs.

### A.4.2 VISION-LANGUAGE ALIGNMENT

Traditional vision-language models (VLMs) have primarily aimed to enhance image-text alignment using methods such as the co-attention framework (Lu et al., 2019), anchor points (Li et al., 2020), and contrastive learning (Radford et al., 2021b). With the significant advancements in large language models (LLMs), recent approaches have explored novel directions to integrate visual encoders with LLMs, enabling better comprehension of vision-language multi-modal tasks. Aligning visual and linguistic modalities can primarily be categorized into two approaches: alignment from training data and alignment from feedback. Alignment from training data involves using high-quality datasets for SFT (Supervised Fine-Tuning) training, including diverse instructions and dataset compression. This method relies on the diversity and quality of the training data to improve the model's performance. Alignment from feedback focuses on fine-tuning the model using feedback of human (Sun et al., 2023; Yu et al., 2023a) or other models like CLIP (Zhou et al., 2024a) and large models (Li et al., 2023d; Zhao et al., 2023). Two primary methods for learning from feedback in VLLMs are Proximal Policy Optimization (PPO) (Sun et al., 2023) and Direct Preference Optimization (DPO) (Zhao et al., 2023; Li et al., 2023d; Chen et al., 2024a). However, these methods encounter challenges. They may generate out-of-distribution data that fails to significantly enhance the model's performance and entail significant expenses in dataset construction.

## A.5 LIMITATIONS

One limitation of FiSAO is its dependency on the quality and robustness of the pre-trained vision models. If the visual encoder contains inherent biases or inaccuracies, these issues can be propagated through the reward model, potentially affecting the overall alignment process.

## A.6  BROADER IMPACTS

The proposed enhancement in Vision-Language Large Models (VLLMs) through fine-grained policy optimization presents several significant broader impacts across various fields and societal dimensions. FiSAOcontributes to the field of AI by providing a novel approach to self-training without the need for additional data. This can inspire further research into data-efficient training methods, fostering innovation and reducing the environmental impact associated with large-scale data collection and processing. Besides, enhanced vision-language alignment can significantly improve the performance of assistive technologies, such as screen readers and automated transcription services, making digital content more accessible to people with disabilities. This aligns with global efforts to promote inclusivity and equal access to information and technology.

