# OpenReview forum: "Fine-Grained Verifiers: Preference Modeling as Next-token Prediction in Vision-Language Alignment"
_ICLR.cc/2025/Conference — ICLR 2025 Poster_

### Official Review · Reviewer_vDxF · 2024-10-30

**Soundness:** 3
**Presentation:** 2
**Contribution:** 3
**Rating:** 5
**Confidence:** 5

**Summary:**

This paper first demonstrates the advantages of token-level over sentence-level approaches in handling hallucinations. It then utilizes the similarity between each text token and visual features as a reward to achieve a token-level preference optimization. This optimization method employs expert models to make judgments on entity targets, enabling automatic construction of preference data. Ultimately, it achieves a certain level of performance improvement.

**Strengths:**

This paper:
1. analyzes the significant advantages of token-level processing over sentence-level processing in handling hallucinations, laying a solid foundation for subsequent research on token-level rewards.
2. claims a token-level visual reward, which relies on expert models to discriminate entity targets. Although the approach is not novel, the idea is commendable.
3. conducts some simple experiments and analyses to verify the effectiveness of the method.

**Weaknesses:**

1. The paper overuses formula derivations while glossing over specific methodological details, making the overall writing quite confusing. Especially in section 3.2 and equation 5, there’s a lack of necessary logic, making it difficult to understand.
2. I understand that the data construction method mentioned in the article still relies on fuzzy matching with ground truth based on entities. This seems to be a rather manual and primitive way of data construction, potentially introducing significant systematic errors. Have you tried more precise token selection methods?
3. Overall, I find the experiments still insufficient. The experimental baselines are quite outdated, and the tested benchmarks are not comprehensive enough. Particularly, why is there a claim about handling hallucinations, but the final experiments lack an evaluation of hallucinations? Also, why is there almost no improvement on POPE? What’s the reason for this?

**Questions:**

See Weaknesses

---

> ### Author Response · Authors · 2024-11-23
> **Response to Reviewer vDxF**
>
> >Q1 The paper overuses formula derivations while glossing over specific methodological details, making the overall writing quite confusing. Especially in section 3.2 and equation 5, there’s a lack of necessary logic, making it difficult to understand.
>
> **A1**
>  Thank you for your valuable feedback and for pointing out the specific issues with Section 3.2 and Equation 5.
> We understand that an overemphasis on formula derivations, without sufficient explanation of the underlying methodology, can make the writing less accessible. To address this concern, we have made the following improvements in the revision version:
> * Improved Logical Flow:  We have revised Section 3.2 to better articulate the connection between the equations and the methodological details, ensuring that each step in the derivation is accompanied by an intuitive explanation. This makes it clearer how the formulas contribute to the proposed approach.
> * Clarity for Equation 5: We have expanded the discussion surrounding Equation 5 to include the rationale and context behind its formulation.
>
>
> >Q2  I understand that the data construction method mentioned in the article still relies on fuzzy matching with ground truth based on entities. This seems to be a rather manual and primitive way of data construction, potentially introducing significant systematic errors. Have you tried more precise token selection methods?
>
> **A2** We believe there may be a misunderstanding regarding our methodology. Could you clarify what specific concerns or aspects you are referring to? I’d like to emphasize that we do not manually construct the data, nor do we rely on fuzzy matching with ground truth entities. Actually, we do not mention data construction in our paper. If there are specific concerns or aspects we might have overlooked, we would greatly appreciate further clarification to address your concerns comprehensively.
>
> >Q3  Overall, I find the experiments still insufficient. The experimental baselines are quite outdated, and the tested benchmarks are not comprehensive enough. Particularly, why is there a claim about handling hallucinations, but the final experiments lack an evaluation of hallucinations? Also, why is there almost no improvement on POPE? What’s the reason for this?
>
> **A3**    Thank you for your valuable feedback on our work. We have carefully addressed your concerns as follows:
> * Updated Baselines:
> We sincerely appreciate your suggestion to include more recent baselines. Following your feedback, we have added the latest baselines, including HA-DPO and RLAIF-V, in Table R1 below. As demonstrated in the updated results, our method exhibits competitive performance compared to these state-of-the-art approaches.
> * Comprehensive Benchmarks:
> Thank you for pointing out the potential confusion in our benchmark setup. To address this, we have ensured our setup strictly follows the settings from LLaVA, encompassing a broad range of evaluation datasets. Specifically, the last three evaluation metrics in Table 2, such as CHAIR_S, CHAIR_I, and COCO_Caption, focus on assessing hallucination. To make this clearer, we have updated the table titles and explicitly noted that a smaller CHAIR value indicates better performance.
> * Concern on POPE:
> We appreciate your careful observation regarding the performance on the POPE dataset. This dataset predominantly consists of simple QA tasks with single-word ground truth answers. In such cases, the difference between sentence-level and token-level rewards is minimal, limiting the advantage of our fine-grained approach. We acknowledge this findings and have included this experimental discussion in our revision.
>
> **Table R1: Performance Comparison Across Baselines**
> | Method           | MME^P| MME^C | SEED | MMB  | MM-Vet | SQA | POPE | GQA  | Cap_val | CHAIR$_S$ | CHAIR$_I$|
> |-------------------|-----------------|-----------------|------|------|--------|-----------------|------|------|---------|-------------------|-------------------|
> | LLaVA-1.5    | 1510.7          | 348.2           | 58.6 | 64.3 | 30.5   | 66.8           | **85.9** | **62.0** | 56.6    | 54.3             | 11.3             |
> | +    HA-DPO  | 1514.3          | 350.4         | 59.5 | 64.4 | 30.5| 68.5           | 85.8 | 61.9| 56.4   | 44.1             | 10.2             |
> | + RLAIF-V     | 1508.2          | 349.1          | 60.1 | 63.4 | 30.5   | 68.4           | 81.5 | 61.9 | 57.6   | 43.2             | 10.3             |
> |**+ Fisao**     | **1522.6**      | **349.0**       | **60.6**| **64.8** | **30.7**| **69.3**        | 85.7 | 62.0 | **61.2**| **39.9**             | **9.9**              |

---

> ### Author Response · Authors · 2024-11-25
> **Follow-up on Review Feedback**
>
> Thank you for your valuable insights. We have improved Section 3.2 and Equation 5 by enhancing logical clarity and providing detailed explanations. We added recent baselines (HA-DPO, RLAIF-V) and clarified hallucination-related benchmarks (e.g., CHAIR).  Results in Table R1 underscore our method’s competitive performance. We look forward to your valuable feedback.

---

### Official Review · Reviewer_ZUH4 · 2024-11-03

**Soundness:** 3
**Presentation:** 2
**Contribution:** 3
**Rating:** 6
**Confidence:** 4

**Summary:**

The paper proposes Fine-Grained Self-Alignment Optimization (FiSAO) to improve vision-language alignment in VLMs. The core idea is to leverage token-level, fine-grained feedback from the model's own visual encoder to guide the training process. The method avoids the need for external annotations or tools by utilizing the vision encoder as a fine-grained verifier, providing token-level rewards during training.

**Strengths:**

- The paper addresses a critical challenge in morden VLMs. The approach of token-level, fine-grained rewards to improve alignment make much sense to me.

- The paper presents a theoretical framework, including mathematical proofs.

- The experimental evaluation is thorough and well-designed. The experiments support the benefits of token-level rewards in reducing hallucinations and improving benchmark performance.

**Weaknesses:**

1. The theoretical analysis assumes that the CLIP provides perfect vision-langauge alignment, which is a bit unrealistic as CLIP models are known to have alignment errors in practice. Under this assumption, it’s unsurprising that it can be proved that incorporating the CLIPScore improves the performance. Also, the presented theorem cannot support the benifit of _fine-grained_ reward, which is the core point of this paper. Additionally, the framework also relies on linear relationships between inputs and outputs and the infinite data setting.

2.  The introduction of the MDP perspective seems unnecessary and is not well-justified within the context of the paper. Framing language generation as an MDP may not capture the long-range dependencies and context inherent in language, potentially violating the Markov property. At the end of Section 3.3.1, the author state that “This perspective highlights how fine-grained rewards can be applied ...”--I am unsure about how introducing the MDP framework demonstrates this point.

3. Regarding Section 3.1

   Overall, this section looks a bit unclear to me. Firstly, the calculation process of token-level reward is not introduced, leading to confusion. Whether it is token-level CLIPScore? (additional question: whether per-sentence average of token-level reward is the same as sentence-level reward?)

   The authors state that sentence-level rewards show a weak correlation with conventional metrics, implying that sentence-level rewards are not effective. However, metrics like BLEU and ROUGE are known to have limitations in measuring semantic similarity, consider using CiDEr or SBERT semantic simialrity score can make the results more convincing. Also, the paper does not clarify whether token-level rewards have a stronger correlation.

5. Issues with Clarity and Readibility

   - Symbols like $S$ in Equation (7) and the normalization function \( \mathcal{N} \) are not properly defined when first introduced. The S also overlapps with the S in MDP.

   - Figure 1 shows distributions of two types of rewards before the concept and calculation of ‘reward’ is introduced.

**Questions:**

N/A

---

> ### Author Response · Authors · 2024-11-23
> **Response to Reviewer ZUH4 (1/2)**
>
> >Q1 The theoretical analysis assumes that the CLIP provides perfect vision-langauge alignment, which is a bit unrealistic as CLIP models are known to have alignment errors in practice. Under this assumption, it’s unsurprising that it can be proved that incorporating the CLIPScore improves the performance. Also, the presented theorem cannot support the benifit of fine-grained reward, which is the core point of this paper. Additionally, the framework also relies on linear relationships between inputs and outputs and the infinite data setting.
>
> **A1** Thank you very much for your thoughtful feedback and for highlighting this crucial point. We sincerely appreciate your attention to detail. You are absolutely correct that the assumption of perfect vision-language alignment in CLIP is not entirely realistic, as alignment errors are well-documented in practice. This simplification, however, is widely adopted in related studies [1, 2] to enable a more manageable theoretical analysis.
> Without this assumption, a rigorous proof becomes significantly more challenging. However, this assumption provides a tractable starting point to analyze the impact of vision feedback on VLLM performance.
> We acknowledge your observation that the presented theorem does not directly support the benefits of fine-grained rewards, which is a core focus of this paper. To address this, **we have included a new proof in Appendix A.3.2  in the revised version** that explicitly demonstrates how token-level rewards contribute to enhanced performance, providing stronger theoretical support for our claims.
>
> [1]  Nakada, R., Gulluk, H. I., Deng, Z., Ji, W., Zou, J., & Zhang, L. (2023, April). Understanding multimodal contrastive learning and incorporating unpaired data. In International Conference on Artificial Intelligence and Statistics (pp. 4348-4380). PMLR.
>
> [2] Zhou, Y., Fan, Z., Cheng, D., Yang, S., Chen, Z., Cui, C., ... & Yao, H. (2024). Calibrated self-rewarding vision language models. arXiv preprint arXiv:2405.14622.
>
>
>
>
> >Q2 The introduction of the MDP perspective seems unnecessary and is not well-justified within the context of the paper. Framing language generation as an MDP may not capture the long-range dependencies and context inherent in language, potentially violating the Markov property. At the end of Section 3.3.1, the author state that “This perspective highlights how fine-grained rewards can be applied ...”--I am unsure about how introducing the MDP framework demonstrates this point.
> >
> **A2**   Thank you for your insightful feedback!
>  To address your concerns, we present the point-to-point response as follows.
>
> **The introduction of the MDP perspective seems unnecessary and is not well-justified within the context of the paper.**
>
> Thanks for your insightful feedback to help us improve clarity of our presentation.  Introducing the MDP framework is crucial for our method because it provides a formal structure to apply fine-grained (token-level) rewards during the generation process. By modeling the VLLM as an agent in an MDP, we can:
> * Define States and Actions: The states represent the current context or history $y_{<t}$, and the actions are the possible next tokens $y_t$. This precise definition is essential for applying RL algorithms that operate on state-action pairs.
> * Apply Rewards at Each Decision Point: The MDP framework allows us to assign rewards at each time step t, corresponding to each token generated. This is particularly important for our fine-grained feedback mechanism, where we want to encourage or discourage specific token-level outputs based on the alignment with the visual content.
> * Leverage RL Algorithms Effectively: By framing the problem within an MDP, we can employ standard RL optimization techniques like PPO to adjust the policy based on the rewards received. This facilitates stable and efficient training.
>
> **Framing language generation as an MDP may not capture the long-range dependencies and context inherent in language, potentially violating the Markov property.**
>
> We appreciate your thoughtful observation regarding the limitations of framing language generation as a Markov Decision Process (MDP), especially concerning the capture of long-range dependencies and context inherent in language, which might indeed challenge the Markov property.
> However,  we believe our approach holds considerable merit. Recent developments suggest that MDP frameworks can be refined and extended to effectively account for long-range dependencies in language generation, offering a robust foundation for modeling such complexities[1,2,3].
>
> [1] Liu G, Ji K, Zheng R, et al. Enhancing Multi-Step Reasoning Abilities of Language Models through Direct Q-Function Optimization[J].
>
> [2] Cao Y, Sheng Q Z, McAuley J, et al. Reinforcement learning for generative ai: A survey[J].
>
> [3] Wu J, Ning L, Liu L, et al. RLPF: Reinforcement Learning from Prediction Feedback for User Summarization with LLMs[J].

---

> ### Author Response · Authors · 2024-11-23
> **Response to Reviewer ZUH4 (2/2)**
>
> >Q3 Regarding Section 3.1
> Overall, this section looks a bit unclear to me. Firstly, the calculation process of token-level reward is not introduced, leading to confusion. Whether it is token-level CLIPScore? (additional question: whether per-sentence average of token-level reward is the same as sentence-level reward?)
> The authors state that sentence-level rewards show a weak correlation with conventional metrics, implying that sentence-level rewards are not effective. However, metrics like BLEU and ROUGE are known to have limitations in measuring semantic similarity, consider using CiDEr or SBERT semantic similarity score can make the results more convincing. Also, the paper does not clarify whether token-level rewards have a stronger correlation.
>
> **A3** Thanks for your response.
> We will address your concern  individually.
>
>
> **Overall, this section looks a bit unclear to me. Firstly, the calculation process of token-level reward is not introduced, leading to confusion. Whether it is token-level CLIPScore? (additional question: whether per-sentence average of token-level reward is the same as sentence-level reward?)**
>
>
> Thank you for pointing out this issue and for highlighting the need for clarity in our explanation of the token-level reward calculation process. We sincerely appreciate your detailed feedback.
> **In the Section 3.1 of revised version, we have clarified that token-level rewards are obtained by calculating the similarity between the token-level text embeddings and the image embeddings from the VLLM’s vision encoder.** In contrast, sentence-level rewards are computed based on the similarity between the embedding of the entire sentence and the image embedding.
>
>
> **The authors state that sentence-level rewards show a weak correlation with conventional metrics, implying that sentence-level rewards are not effective. However, metrics like BLEU and ROUGE are known to have limitations in measuring semantic similarity, consider using CiDEr or SBERT semantic similarity score can make the results more convincing.**
>
>
> Thank you for your valuable feedback and for suggesting the use of alternative metrics such as CiDEr and SBERT semantic similarity score to strengthen our evaluation. We appreciate your observation regarding the limitations of BLEU and ROUGE in measuring semantic similarity.
> We have conducted additional experiments using metrics such as CiDEr and Meteor   score to provide a more comprehensive evaluation. These results, originally presented in Figures 8 and 9 of the Appendix, **have been moved to Figure 2 in  the main text** to address the reviewer's concern.
>
>
> **Also, the paper does not clarify whether token-level rewards have a stronger correlation.**
>
> To address the reviewer's concern, we calculate the average sum of token-level rewards in a sentence and explore its relationship with conventional evaluation metrics at the sentence level. **As shown in Figure 10, we compare the correlation between sum of token-level rewards and conventional evaluation metrics with the correlation between sentence-level rewards and conventional evaluation metrics.** We observe that, compared to sentence-level rewards, token-level rewards exhibit strong correlation with conventional evaluation metrics.
>
> >Q4 Issues with Clarity and Readibility
>
> **A4**  Thank you for your feedback. **We have revised the manuscript to clearly define symbols  like  $S$ and $\mathcal{N}$ in Section 3.3.2 when first introduced, and restructured the section 3.1 so that reward concepts are introduced before Figure 1.**

---

> ### Author Response · Authors · 2024-11-25
> **Follow-up on Review Feedback**
>
> Thank you for your thoughtful comments. We have clarified token-level reward calculations and their distinction from sentence-level rewards (Section 3.1), incorporated CiDEr and SBERT for better semantic evaluation, and improved clarity of Section 3.3.2 symbols and structure. Results now show stronger token-level reward correlations (Figure 10). We look forward to your feedback.

---

### Official Review · Reviewer_iK73 · 2024-11-04

**Soundness:** 3
**Presentation:** 3
**Contribution:** 3
**Rating:** 6
**Confidence:** 4

**Summary:**

The paper presents Fine-Grained Self-Alignment Optimization (FiSAO) to improve the alignment of visual and linguistic modalities in vision-language large models (VLLMs). It addresses the challenges of misalignment caused by independent pre-training, which can lead to biased outputs and hallucinations. By leveraging token-level feedback from the vision encoder, FiSAO enhances alignment more effectively than traditional methods that rely on coarse feedback. The findings suggest that this approach reduces the need for external data and improves performance in tasks requiring precise integration of visual and language information.

**Strengths:**

- The paper is well-structured and clearly articulated, making it accessible for readers.
- The introduction of FiSAO is grounded in sound analysis and presents a logical approach to addressing alignment issues.
- Extensive experimental validation supports the effectiveness of FiSAO, showcasing its practical applicability.
- The authors offer valuable insights into enhancing alignment in vision-language large models, contributing meaningfully to the field.

**Weaknesses:**

- In the comparison with other state-of-the-art methods, FiSAO did not achieve results on MM-Vet that are comparable to leading approaches. It would be beneficial to include an analysis of the factors contributing to this outcome and to outline potential strategies for improvement in future iterations. Understanding the specific challenges faced in this context could guide refinements to FiSAO and the design of models.
- Efficiency of FiSAO should be analysed as well.  And how the dependency of pretrained vision model affects the performance?

**Questions:**

Please see above.

---

> ### Author Response · Authors · 2024-11-23
> **Response to Reviewer iK73 (1/2)**
>
> >Q1 In the comparison with other state-of-the-art methods, FiSAO did not achieve results on MM-Vet that are comparable to leading approaches. It would be beneficial to include an analysis of the factors contributing to this outcome and to outline potential strategies for improvement in future iterations. Understanding the specific challenges faced in this context could guide refinements to FiSAO and the design of models.
>
>  **A1** Thank you for your thoughtful feedback and for highlighting this important observation. We greatly appreciate your recognition of the need to analyze FiSAO's performance on MM-Vet, as this helps us identify areas where our approach can be refined.
>  MM-Vet primarily involves simple QA tasks, where the ground truth is often a single word, making it difficult to leverage the advantages of token-level rewards.  We believe this represents an interesting direction for future exploration.

---

> ### Author Response · Authors · 2024-11-23
> **Response to Reviewer iK73 (2/2)**
>
> >Q2 Efficiency of FiSAO should be analysed as well. And how the dependency of pretrained vision model affects the performance?
>
> **A2**  Thank you for your valuable feedback! We have conducted experiments to analyze the efficiency of FiSAO, with the detailed results provided in Table R1. **A comprehensive analysis of these findings can be found in Appendix A.2.7 in the revised version.**
>
> **Table R1: Computational Efficiency Across Methods**
> | Method                | Time per Sample (seconds) |  Sample per Second|
> |------------------------|---------------------------|-----------------------------|
> | **Inference Only**     | 2.63                     | 0.38                        |
> | **POVID**              | 2.15                     | 0.47                        |
> | **VLFeedback**         | 2.34                     | 0.43                        |
> | **Fisao (w/o Inference)** | 1.17  | 0.85                        |
> | **Fisao (w Inference)**  |  3.80    | 0.26                        |
>
> When excluding inference time, our method is more efficient than others, with a faster reward computation process. For the overall time (including inference), the efficiency of our approach is comparable to POVID and VLFeedback. However, our method delivers better performance due to the use of fine-grained token-level rewards, highlighting its practical advantages despite similar overall efficiency.
>
>
>
> Additionally, we sincerely appreciate you bringing attention to the impact of the pre-trained vision model on FiSAO's performance.  To explore this,  we conduct two experiments:
>
> * a:  We adopt DINO-v2 as the reward model to provide fine-grained feedback for training LLaVA. From Table R2, it is evident that our method (Fisao (Dino)) achieves superior performance across multiple metrics, such as the highest MME$^{\mathrm{P}}$, MME$^{\mathrm{C}}$, and SEED values, while significantly reducing CHAIR${\mathrm{S}}$ and CHAIR${\mathrm{I}}$.
> **The corresponding analyses and results are included in the revised version under Section A.2.5.**
>
> **Table R2: Performance Comparison Across Baselines**
> | **Method**         | **MME$^{\mathrm{P}}\uparrow$** | **MME$^{\mathrm{C}}\uparrow$** | **SEED $\uparrow$** | **Cap\_val $\uparrow$** | **CHAIR$_{\mathrm{S}}$ $\downarrow$** | **CHAIR$_{\mathrm{I}}$ $\downarrow$** | **SciQA $\uparrow$** |
> |---------------------|--------------------------------|--------------------------------|---------------------|--------------------------|---------------------------------------|---------------------------------------|---------------------|
> | **LLaVA-1.5**       | 1510.7                        | 348.2                         | 58.6                | 56.6                    | 54.3                                  | 11.3                                  | 66.8                |
> | **+ Vlfeedback**    | 1432.7                        | 321.8                         | 59.3                | 54.8                    | 40.3                                  | 13.2                                  | 66.2                |
> | **+ Human-Prefer**  | 1490.6                        | 335.0                         | 58.1                | 50.4                    | 38.7                                  | 11.3                                  | 65.8                |
> | **+ POVID**         | 1452.8                        | 325.3                         | 60.2                | 57.3                    | **35.2**                                  | **8.3**                               | **68.8**                |
> | **+ Fisao (Dino)**        | **1542.6**                    | **351.1**                     | **60.3**            | **61.5**                | 37.4                              | 9.3                                   | 68.7           |
>
>
>
>
> * b:  Due to the time limitation we are unable to train Fisao using SigLip scores as rewards. However, to address the reviewer's concern, we visualized the distributions of hallucinated and correct tokens using SigLip scores, as shown in Figure 12.** A detailed analysis has been included in the revised version in Appendix A.2.5.**   We observe that SigLip demonstrates better differentiation between hallucinated and correct tokens compared to CLIP. This suggests that our approach is potentially highly adaptable when applied to MLLMs incorporating multiple visual encoders.
> Regarding the modifications or considerations necessary to handle such configurations effectively, the advantage of our method is its simplicity and flexibility. By following the reward calculation approach in Equation (8), our framework can seamlessly incorporate feedback from any visual encoder, including SigLip or others. This adaptability makes our method versatile and effective across various settings and encoder architectures.

---

> ### Author Response · Authors · 2024-11-25
> **Follow-up on Review Feedback**
>
> We appreciate your detailed feedback. We have analyzed FiSAO's efficiency and provided computational overhead results (Table R1). We also explored the dependency on pre-trained vision models with DINO-v2 and SigLip (Appendix A.2.5). These findings underscore FiSAO's adaptability and effectiveness in diverse multimodal configurations. We look forward to your further insights.

---

### Official Review · Reviewer_SJcY · 2024-11-08

**Soundness:** 3
**Presentation:** 3
**Contribution:** 3
**Rating:** 8
**Confidence:** 4

**Summary:**

This paper introduces Fine-Grained Self-Alignment Optimization (FiSAO), a method designed to improve vision-language alignment in Vision-Language Large Models (VLLMs). Unlike traditional alignment techniques that use coarse, sentence-level feedback, FiSAO leverages token-level feedback from the model’s visual encoder, requiring no additional external data or annotations. This token-level approach addresses hallucination issues with more precise differentiation between hallucinated and correctly grounded outputs.

**Strengths:**

- Token-level rewards is an emerging area of research that is worth exploring. So far, only contemporary work has begun to consider token-level reward methods for VLMs, with this paper being one of the first works in this space.
- The methodological framework is well-defined, with both empirical and theoretical foundations.
- Evaluation is comprehensive, with many baselines and benchmarks.

**Weaknesses:**

- Table 1 does not seem to be very representative of relevant literature. Neither the motivation nor the related work covers existing methods for NLP token-level reward [1,2]. Would be nice to also cover concurrent works for VLMs [3,4] so that differences with recent work are better highlighted (albeit not taken into consideration as a weakness in my review, to the contrary, this shows this area of research is very important).

- It is unclear why not use OpenCHAIR which addresses the limitations of CHAIR for benchmarking object hallucinations.

- Figure 2 alone does not robustly support the observation that sentence-level rewards are unreliable indicators of model performance. To be exact, Figure 2 suggests there exists no *linear* relationship between the CLIP-based sentence rewards and the conversational metrics. Sentence-level CLIP scores might still have value, but in ways not captured by simple linear correlation with BLEU or ROUGE. Perhaps the figure could also show the linear correlation for token-level rewards to make a comparative argument, which would be stronger.

- Theoretical analysis assumes linear transformations and relationships between the latent representations of the image and text modalities (e.g., in the data generative model for v and t). It also relies on orthogonal matrices U_v and U_t​ for projecting latent variables to high-dimensional representations. While this simplifies the theoretical analysis, it assumes a degree of independence or decorrelation between features that may not exist in complex multimodal settings. There are other simplifications such as sub-Gaussian noise, Gaussian generative likelihood, an infinite data setting, and modeling token-level feedback as a regression problem, all of which understandably simplify (and make possible) the analysis but perhaps should be stated clearly as assumptions.

- The current method relies on predefined sets of objects and labels (from datasets like Detic and COCO) to define common objects for fine-grained reward calculation and the fine-grained reward calculation uses predefined thresholds to distinguish correct and hallucinated tokens. Seems both would require careful tuning.

- Calculating token-level rewards for every token in a generated sequence seems computationally intensive? The paper contains no analysis of the performance - computational overhead tradeoffs against sentence-level rewards.

- The reward in Eq. (8) is too difficult to parse, perhaps consider improving clarity in the draft notation presentation format.

- Please explain what is POVID, etc., in the main paper? The appendix does not seem to be a good place to add this information since these methods are crucial in understanding the results. Also briefly mention what are the eval metrics used for each Table 2 column? I could not locate this information easily.

- Also, it is unclear if Table 2 shows results where FiSAO is added on top of other methods such as POVID, Human-Prefer, etc. Would it make more sense to evaluate the method with just RLHF sentence-level or token-level feedback to show clear differences?

[1] Yoon, Eunseop, Hee Suk Yoon, SooHwan Eom, Gunsoo Han, Daniel Nam, Daejin Jo, Kyoung-Woon On, Mark Hasegawa-Johnson, Sungwoong Kim, and Chang Yoo. "TLCR: Token-Level Continuous Reward for Fine-grained Reinforcement Learning from Human Feedback." In Findings of the Association for Computational Linguistics ACL 2024, pp. 14969-14981. 2024.

[2] Yang, Kailai, Zhiwei Liu, Qianqian Xie, Jimin Huang, Erxue Min, and Sophia Ananiadou. "Selective Preference Optimization via Token-Level Reward Function Estimation." arXiv preprint arXiv:2408.13518 (2024).

[3] Fu, Deqing, Tong Xiao, Rui Wang, Wang Zhu, Pengchuan Zhang, Guan Pang, Robin Jia, and Lawrence Chen. "TLDR: Token-Level Detective Reward Model for Large Vision Language Models." arXiv preprint arXiv:2410.04734 (2024). **[Concurrent work]**

[4] Xu, Yuancheng, Udari Madhushani Sehwag, Alec Koppel, Sicheng Zhu, Bang An, Furong Huang, and Sumitra Ganesh. "GenARM: Reward Guided Generation with Autoregressive Reward Model for Test-time Alignment." arXiv preprint arXiv:2410.08193 (2024). **[Concurrent work]**

**Questions:**

- What are the major differences between this work and recent token-level reward papers in NLP and VLMs?

- What is the correlation between CLIP-based **token-level** rewards and conversational metrics?

-  What are the eval. metrics used for each Table 2 column?

-  To my understanding, Table 2 shows results where FiSAO is added **on top of** other methods such as POVID, Human-Prefer, etc.? Would it make more sense to evaluate the method with just RLHF sentence-level or token-level feedback to show clear differences?

I find the work interesting and timely, and look forward to the rebuttal.

---

> ### Author Response · Authors · 2024-11-23
> **Response to Reviewer SJcY (1/2)**
>
> >Q1 Table 1 does not seem to be very representative of relevant literature. Neither the motivation nor the related work covers existing methods for NLP token-level reward [1,2]. Would be nice to also cover concurrent works for VLMs [3,4] so that differences with recent work are better highlighted (albeit not taken into consideration as a weakness in my review, to the contrary, this shows this area of research is very important).
>
> **A1** Thank you for your constructive feedback.
> In response, we have expanded  **related work sections in Appendix A.4.3**  to discuss these references, highlighting the differences and contributions of our approach more comprehensively. We greatly appreciate your insightful comments, which have helped us improve the quality of our work.
>
> >Q2  It is unclear why not use OpenCHAIR which addresses the limitations of CHAIR for benchmarking object hallucinations.
>
> **A2**  Thank you for your valuable feedback. We have  incorporated additional experiments using OpenCHAIR to benchmark object hallucinations. The results have been included in Table R1. We can observe that Fisao achieves the lowest CH$_i$ (9.6) and CH$_s$ (45.6), outperforming other baselines in reducing hallucinations.
>
> **Table R1: Performance on Object Hallucination Benchmarks (OpenCHAIR)**
>
> | Model     | **OCH↓** |
> |-------------|----------|
> | LLaVA  + vlfeedback     | 0.387|
> | LLaVA + human-prefer  | 0.402 |
> | LLaVA + POVID     | 0.393    |
> |**LLaVA + Fisao**     | 0.379  |
>
>
> >Q3  Figure 2 alone does not robustly support the observation that sentence-level rewards are unreliable indicators of model performance. To be exact, Figure 2 suggests there exists no linear relationship between the CLIP-based sentence rewards and the conversational metrics. Sentence-level CLIP scores might still have value, but in ways not captured by simple linear correlation with BLEU or ROUGE. Perhaps the figure could also show the linear correlation for token-level rewards to make a comparative argument, which would be stronger.
>
> **A3**  Thank you for your thoughtful and detailed feedback! We greatly appreciate your attention to the nuances of our analysis and your suggestion to strengthen the comparative argument by including linear correlation for token-level rewards.
>
> To address the reviewer's concern, we calculate the sum of token-level rewards in a sentence and explore its relationship with conventional evaluation metrics at the sentence level.  **We have made corresponding modifications and clarifications in the revised version.**  In Appendix 2.2.2, we compare the correlation between sum of token-level rewards and conventional evaluation metrics with the correlation between sentence-level rewards and conventional evaluation metrics. As shown in Figure 9, we observe that, compared to sentence-level rewards, token-level rewards exhibit strong correlation with conventional evaluation metrics.
>
>
> >Q4  Theoretical analysis assumes linear transformations and relationships between the latent representations of the image and text modalities (e.g., in the data generative model for v and t). It also relies on orthogonal matrices U_v and U_t​ for projecting latent variables to high-dimensional representations. While this simplifies the theoretical analysis, it assumes a degree of independence or decorrelation between features that may not exist in complex multimodal settings. There are other simplifications such as sub-Gaussian noise, Gaussian generative likelihood, an infinite data setting, and modeling token-level feedback as a regression problem, all of which understandably simplify (and make possible) the analysis but perhaps should be stated clearly as assumptions.
>
>
> **A4**  Thank you for your detailed and perceptive feedback! We greatly appreciate your careful examination of our theoretical analysis and your thoughtful suggestions for clearly stating the assumptions made.
> In response to your observations, **we have explicitly outlined these assumptions in the revised Section 3.2**, ensuring that the degree of independence or decorrelation between features, the sub-Gaussian noise, and Gaussian generative likelihood are clearly acknowledged as necessary conditions for simplifying the analysis.

---

> > ### Author Response · Authors · 2024-11-23
> > **Response to Reviewer SJcY (2/2)**
> >
> > >Q5 Calculating token-level rewards for every token in a generated sequence seems computationally intensive? The paper contains no analysis of the performance - computational overhead tradeoffs against sentence-level rewards.
> >
> >
> > **A5**   Thank you for your insightful comment.
> > Token-level rewards and sentence-level rewards exhibit roughly comparable computational overhead. While sentence-level rewards reduce the number of reward calculations, the longer token sequences involved slow down the reward computation process.
> > To substantiate our point, we conduct experiments comparing the two approaches. As detailed in Table R2,  the difference in  computation time per sample  is not significant, suggesting practical feasibility for most applications.
> >  **In the revised version of our paper, we have added a detailed comparison of the computational overhead for token-level and sentence-level rewards in Appendix A.2.7.**
> >
> > **Table R2: Computational Overhead Comparison Between Token-Level and Sentence-Level Rewards**
> >
> > | Metric               | Token-Level                 | Sentence-Level                |
> > |-----------------------|-----------------------------|-------------------------------|
> > | Total Samples         | 5000                       | 5000                          |
> > | Current Speed (s/sample) |  0.1883                    |  0.1631                       |
> >
> >
> >
> > Our experiments indicate that token-level rewards do not significantly impact efficiency. To further substantiate this, we also compare the total runtime across different methods, presented in Table R3:
> >
> > **Table R3: Comparison of Computational Overhead for Different Methods**
> >
> > | Method                | Time per Sample (seconds) | Throughput (samples/second) |
> > |------------------------|---------------------------|-----------------------------|
> > | Inference Only     | 2.63                     | 0.38                        |
> > | POVID              | 2.15                     | 0.47                        |
> > | VLFeedback         | 2.34                     | 0.43                        |
> > | Fisao(w/o Inference) | 1.17                   | 0.85                        |
> > | Fisao(w Inference)  | 3.80                     | 0.26                        |
> >
> > When excluding inference time, our method demonstrates higher efficiency with faster reward computation compared to others. For overall runtime (including inference), our efficiency is comparable to POVID and VLFeedback.
> >
> > >Q6 The reward in Eq. (8) is too difficult to parse, perhaps consider improving clarity in the draft notation presentation format.
> >
> > **A6**  Thank you for your feedback. **In the revised version, we have improved its presentation format of Eq. (8) to make it more intuitive and easier to parse.** Key terms are now briefly explained within the main text for clarity.
> >
> >
> >
> >
> > >Q7 Please explain what is POVID, etc., in the main paper? The appendix does not seem to be a good place to add this information since these methods are crucial in understanding the results. Also briefly mention what are the eval metrics used for each Table 2 column? I could not locate this information easily.
> >
> > **A7** Thank you for your valuable suggestions and for highlighting this important aspect. **In the revised version, we have moved the descriptions of POVID and other methods from the appendix to the main paper  in Section 4.1**, as they are critical for understanding the results.
> > **Additionally, we have included clear explanations of the evaluation metrics used for each column in Table 2  within the main text to make this information more accessible.** We greatly appreciate your feedback, which has helped us improve the clarity and readability of our work.
> >
> >
> > >Q8 Also, it is unclear if Table 2 shows results where FiSAO is added on top of other methods such as POVID, Human-Prefer, etc. Would it make more sense to evaluate the method with just RLHF sentence-level or token-level feedback to show clear differences?
> > >
> > **A8** Thank you for bringing this to our attention.   We sincerely appreciate your observation. To clarify, Table 2 does not present results where FiSAO is added on top of other methods such as POVID or Human-Prefer. Instead, these methods serve as baselines for comparison. Our evaluation focuses on assessing FiSAO independently against these baselines to demonstrate its performance.
> > **We  have  stated this distinction  explicitly  in the revised Section 4.1 to avoid any confusion.**

---

> ### Author Response · Authors · 2024-11-25
> **Follow-up on Review Feedback**
>
> Thank you for your constructive comments. We have thoroughly addressed your concerns and added comprehensive discussions on related works (Appendix A.4.3), benchmarking with OpenCHAIR (Table R1), and clarified assumptions in theoretical analysis (Section 3.2). Detailed improvements are reflected in Figures 9, 12, and Tables R2-R3. We look forward to your further feedback.

---

> > ### Comment · Reviewer_SJcY · 2024-11-29
> > **Thorough rebuttal but marginal improvements**
> >
> > Thank you for the rebuttal. The paper revisions expand on related work, limitations, OpenChair comparisons, clarity of the theoretical analysis, etc. However, I am not sure what Fig. 12 tries to capture, as it seems to have a different format than Fig. 2 and there seems to be limited discussion on this aspect. Also, given Table R3, it does not seem that efficiency is comparable to POVID and VLFeedback as the rebuttal mentions. Finally, the experimental results show marginal improvements over existing methods, e.g., POVID.

---

> > > ### Author Response · Authors · 2024-12-02
> > > **Response to Reviewer SJcY**
> > >
> > > Apologies for the delayed response, and thank you for your insightful comments and feedback. Below, we address your concerns regarding Figures, efficiency comparisons, and experimental results in detail.
> > >
> > > > Q1 However, I am not sure what Fig. 12 tries to capture, as it seems to have a different format than Fig. 2 and there seems to be limited discussion on this aspect.
> > > >
> > > **A1** We sincerely apologize for the confusion caused. The relationship between token-level rewards and conventional evaluation methods is shown in Fig. 9. We have corrected this in the revised version.
> > >
> > > **Limited discussion:** A limitation of our analysis is that conventional metrics are sentence-level evaluations, while our study uses token-level rewards. As a result, we approximate their relationship by summing token-level rewards, which may not fully capture the nuances between them.
> > >
> > > >Q2 Also, given Table R3, it does not seem that efficiency is comparable to POVID and VLFeedback as the rebuttal mentions.
> > >
> > > **A2**  We apologize for the confusion caused by our initial explanation. To clarify, the time for generating samples was not included in the efficiency calculations for POVID and VLFeedback.
> > >
> > > POVID involves using GPT-4V to construct data, which incurs significant financial and time costs. VLfeedback involves generating diverse responses using 12 different LVLMs (e.g., GPT-4V, LLaVA, BLIP, etc.), which is far more costly than simply using the trained model itself to generate responses.
> > >
> > > To address the reviewer’s concerns, we have revised the original table by including detailed annotations to clarify the computational overhead across methods, as shown in **Table R1** below.  It can be observed that when the time for generating samples is excluded, our method demonstrates comparable time efficiency.
> > >
> > > **Table R1: Computational Efficiency Across Methods**
> > >
> > > | **Method**                     | **Time per Sample (seconds)** | **Samples per Second** |
> > > |--------------------------------|------------------------------|-------------------------|
> > > | **POVID** (w/o Inference)      | 2.15                        | 0.47                   |
> > > | **VLFeedback** (w/o Inference) | 2.34                        | 0.43                   |
> > > | **FISAO (w/o Inference)**      | **1.17**                    | **0.85**               |
> > > | **FISAO (w/ Inference)**       | 3.80                        | 0.26                   |
> > >
> > >
> > >
> > > >Q3 Finally, the experimental results show marginal improvements over existing methods, e.g., POVID.
> > > >
> > > **A3**  Thank you for your reply. We calculated that FiSAO offers an average improvement of 3.71% compared to POVID. This demonstrates that FiSAO provides a noticeable boost in performance for LVLMs.
> > >
> > > To further address the reviewer's concerns, we compare the results of POVID using only the second-stage approach, without introducing additional data, to ensure a fair comparison.
> > >
> > > Specifically, in the first stage of POVID, GPT-generated data is introduced to augment the training process. In the second stage of POVID, where noise-based augmentation techniques are employed to enhance the learning process.
> > >
> > > It can be observed that our method achieves improvements across all benchmarks compared to POVID.
> > >
> > > **Table R2  Comparison of Performance between FISAO and POVID**
> > >
> > > | Method               | CHAIR$_S$↓ | CHAIR$_I$↓ | MME^p  | MME^C | SEED | Cap_val | SciQA |
> > > |----------------------|--------------------|---------|--------|-------|------|---------|-------|
> > > | POVID (Second-Stage) | 50.4               | 9.6    | 1449.6 | 331.1 | 58.4 | 56.8    | 67.3  |
> > > | FISAO (Dino)         | 37.4               | 9.3     | 1542.6 | 351.1 | 60.3 | 61.5    | 68.7  |

---

> > > > ### Comment · Reviewer_SJcY · 2024-12-03
> > > > **Thank you for the detailed explanations and revisions**
> > > >
> > > > Thank you for the detailed clarifications. While the token-level reward correlation is indeed significantly higher than the sentence-level reward correlation, it remains moderate. Given the inherent challenges in aggregating token-level rewards, incorporating human evaluation, such as measuring the correlation between rewards (sentence-level vs. token-level) and human judgments, could further strengthen this argument. That said, the additional discussion on the limitations of POVID and VLFeedback has provided valuable context, and I appreciate the authors' efforts to address the reviewers' comments. I have raised my score in acknowledgment of these improvements. However, it might also be beneficial to report TFLOPs and inference time.

---

### Official Review · Reviewer_3Rvt · 2024-11-12

**Soundness:** 3
**Presentation:** 3
**Contribution:** 3
**Rating:** 6
**Confidence:** 3

**Summary:**

This paper addresses the issue of modality misalignment in Vision-Language Language Models (VLLM). The authors demonstrate that a token-level reward is more effective than a sentence-level reward for aligning modalities. They apply a specially designed token-level reward during preference tuning to mitigate modality misalignment.

**Strengths:**

1. The proposed self-training approach does not need any additional data and external modules to mitigate the misalignment issue.
2. They find that coarse feedback, such as sentence-level rewards, shows a weak correlation with hallucination detection.
3. They are the first method to introduce token-level rewards for VLLMs preference tuning.

**Weaknesses:**

See questions.

**Questions:**

1. In Figure 2, you show that the correlation between CLIP-based sentence rewards and conventional evaluation metrics is weak. Could you also present the correlation between token-level rewards and conventional evaluation metrics?

2. In the Table 3 experiment, it would be valuable to see the performance improvements across all baselines when using your approach. While line 429 mentions that the LLaVA backbone with your method surpasses existing approaches, it is worth noting that the original LLaVA-1.5 already performed well compared to other models. Demonstrating consistent improvements across various VLLMs could serve as strong evidence of your approach’s generalizability.

3. It is clear from Table 3 that your approach does not allow LLaVA-1.5 to surpass the higher-performing baselines on the MM-Vet benchmark. Additionally, Table 4 shows that fine-grained rewards do not yield an advantage on some benchmarks, including MME^c, POPE, and CHAIR. I am particularly interested in understanding which types of benchmarks or backbone architectures benefit most from your approach and where it shows superior performance.

4. In Figure 4, you illustrate the comparison of reward distributions for generated objects on LLaVA-1.5 before and after training. It seems reasonable that training the model with an RL algorithm on the same reward function results in higher average rewards. However, I am unclear about the purpose of showing the shift in reward distributions. Could you elaborate on the rationale behind presenting this distribution shift? Additionally, could you provide a detailed explanation for the design of Equation (8) and the reasoning behind its specific form?

5. Currently, new MLLMs leverage not only CLIP-based models but also other visual encoders, such as DINO-v2 (self-supervised) and SigLip. How adaptable is your approach when applied to MLLMs that incorporate multiple visual encoders? Could you explain any modifications or considerations necessary for your method to handle such configurations effectively?

6. Your approach primarily focuses on object hallucination. I am curious to know whether it also helps mitigate other types of hallucinations, such as those related to actions or spatial relationships. Additionally, could you elaborate on how the size of the expanded set C contributes to the observed performance improvements?

---

> ### Author Response · Authors · 2024-11-23
> **Response to Reviewer 3Rvt (1/4)**
>
> > Q1   In Figure 2, you show that the correlation between CLIP-based sentence rewards and conventional evaluation metrics is weak. Could you also present the correlation between token-level rewards and conventional evaluation metrics?
> >
> **A1**:  Thank you for raising this insightful question. We greatly appreciate your question, as it provides valuable insights to improve the presentation of our work.
> In response to your concern,  we calculate the average sum of token-level rewards in a sentence and explore its relationship with conventional evaluation metrics at the sentence level.  **We have added a detailed analysis in the revision  in Appendix A.2.2 and present visualization results in Figure 10.** For the visualization experiments, we follow the same settings described in Section 3.1.   As shown in Figure 10, we observe that, compared to sentence-level rewards, token-level rewards exhibit strong correlation with conventional evaluation metrics.
>
>
> >Q2  In the Table 3 experiment, it would be valuable to see the performance improvements across all baselines when using your approach. While line 429 mentions that the LLaVA backbone with your method surpasses existing approaches, it is worth noting that the original LLaVA-1.5 already performed well compared to other models. Demonstrating consistent improvements across various VLLMs could serve as strong evidence of your approach’s generalizability.
> >
> **A2** Thank you for pointing out this important observation. We agree that demonstrating consistent improvements across various VLLMs is crucial for validating the generalizability of our approach.  **We have incorporated it and added new experimental results for InstructBLIP, presented in Table 3 of the revised version.**  As shown in Table R1, our approach achieves consistent improvements across multiple VLLMs, including LLaVA-1.5 and InstructBLIP, further demonstrating the generalizability and effectiveness of our method.
>
> **Table R1: Performance Comparison Across VLLMs**
>
> | Method               | MME^P↑ | MME^C↑ | SEED ↑ | MMB ↑ | MM-Vet ↑ | SQA ↑ | GQA ↑ |
> |----------------------|-------------------|-------------------|--------|-------|----------|-------------------|-------|
> | BLIP-2              | 1293.8           | 290.0             | 46.4   | 38.1  | 22.4     | 61.0              | 41.0  |
> | Qwen-VL-Chat        | 1487.6           | 360.7             | 58.2   | 60.6  | **47.3** | 68.2              | 57.5  |
> | mPLUG-Owl2          | 1450.2           | 313.2             | 57.8   | 64.5  | 36.2     | 68.7              | 56.1  |
> | LLaVA-1.5       | 1510.7           | 348.2             | 58.6   | 64.3  | 30.5     | 66.8              | 62.0  |
> | **LLaVA-1.5 + Fisao** | **1522.6**       | **349.0**         | **60.6**| **64.8**| 30.7     | **69.3**          | **62.0** |
> | InstructBlip         | 1237.5           | 292.1             | 38.5   | 36.0  | 26.0     | 43.5              | 48.0  |
> | **InstructBlip + Fisao** | 1398.0           | 318.9             | 38.9   | 37.4  | 26.9     | 46.3              | 48.2  |
>
> >Q3 It is clear from Table 3 that your approach does not allow LLaVA-1.5 to surpass the higher-performing baselines on the MM-Vet benchmark. Additionally, Table 4 shows that fine-grained rewards do not yield an advantage on some benchmarks, including MME^c, POPE, and CHAIR. I am particularly interested in understanding which types of benchmarks or backbone architectures benefit most from your approach and where it shows superior performance.
>
>
> **A3**    Thank you for highlighting these critical points and for pointing out areas where our approach has limitations. Your feedback  has helped us improve the clarity of our work.
> Below, we provide our responses to address your concerns, organized based on different benchmarks:
>
> * MM-Vet and POPE:  These data primarily involves simple QA tasks, where the ground truth is often a single word, making it difficult to leverage the advantages of token-level rewards.   This limitation could be further explored in future work, such as incorporating new reward computation methods that extend beyond object-based considerations.
>
> * CHAIR: For CHAIR, as shown in Table 4 in our paper, the performance on CHAIR metric improves after using fine-grained rewards.
> The reviewer’s confusion might stem from the fact that we did not clarify that a smaller CHAIR value indicates better performance.  We apologize for any confusion caused by the lack of clarification regarding the interpretation of CHAIR values.  **We have addressed this issue in the revised version  by adding arrows in all the Tables for clarification.**

---

> ### Author Response · Authors · 2024-11-23
> **Response to Reviewer 3Rvt (2/4)**
>
> >Q4 In Figure 4, you illustrate the comparison of reward distributions for generated objects on LLaVA-1.5 before and after training. It seems reasonable that training the model with an RL algorithm on the same reward function results in higher average rewards. However, I am unclear about the purpose of showing the shift in reward distributions. Could you elaborate on the rationale behind presenting this distribution shift? Additionally, could you provide a detailed explanation for the design of Equation (8) and the reasoning behind its specific form?
> >
> **A4**  Thank you for your insightful question! We greatly appreciate your attention to the rationale behind presenting the shift in reward distributions and the design of Equation (8), as it helps us refine our analysis.
> We will address your comments sentence-by-sentence as below:
>
> **However, I am unclear about the purpose of showing the shift in reward distributions.**
>
> The purpose of showing the shift in reward distributions in Figure 4 is to illustrate how our method improves the alignment between the visual and language modalities of the model. Before training, the model tends to generate objects with lower reward values. This indicates that the outputs are not well-aligned with the preferences encoded in the reward function.
> After training with our method, the reward distribution shifts to the right, meaning that the generated objects have higher average rewards. This shift reflects improved alignment and demonstrates that the model learns to generate objects that are  consistent with the image descriptions. By visualizing this change, we aim to highlight the effectiveness of our approach in enhancing vision-language alignment and producing higher-quality outputs.
>  **In the revised version, we have refined the description in Section  4.3 to ensure the rationale for presenting the distribution shift is explicitly stated.**
>
>
> **Additionally, could you provide a detailed explanation for the design of Equation (8) and the reasoning behind its specific form?**
>
> Equation (8) calculates fine-grained rewards to help the model refine its outputs based on the visual input. The reward function evaluates each generated token's score, which indicates how well the token matches the visual input $v$, against two reference scores: the average score for correct objects $\mu_{\text{gt}}$ and the average score for hallucinated objects $\mu_{\text{hal}}$. These reference scores are obtained from the model's visual encoder.
> To distinguish correct outputs from incorrect ones, a margin $\lambda$ is added to create thresholds. If a token’s score is below $\mu_{\text{hal}} - \lambda$, it gets a negative reward, suggesting it is likely incorrect. If the score is above  $\mu_{\text{gt}} + \lambda$, it gets a positive reward, indicating it aligns well with the visual input. Tokens with scores between these thresholds receive no reward, as their accuracy is uncertain.
> **We have included a detailed explanation of Equation (8)  (Equation (12) in the revised version ) to clarify its specific design and the reasoning behind its form.**

---

> ### Author Response · Authors · 2024-11-23
> **Response to Reviewer 3Rvt (3/4)**
>
> >Q5 Currently, new MLLMs leverage not only CLIP-based models but also other visual encoders, such as DINO-v2 (self-supervised) and SigLip. How adaptable is your approach when applied to MLLMs that incorporate multiple visual encoders? Could you explain any modifications or considerations necessary for your method to handle such configurations effectively?
>
> **A5**  We are deeply grateful for your feedback. It’s excellent that you’ve highlighted the potential for adapting our approach to MLLMs with multiple visual encoders, such as SigLip and DINO-v2. To explore this,  we conduct two experiments.
> * a: We adopt DINO-v2 as the reward model to provide fine-grained feedback for training LLaVA. The results are presented in Table R1, and **corresponding analyses and results are included in the revised version under Appendix A.2.5.**
>
> **Table R1: Performance Comparison Across Baselines**
> | **Method**         | **MME$^{\mathrm{P}}\uparrow$** | **MME$^{\mathrm{C}}\uparrow$** | **SEED $\uparrow$** | **Cap\_val $\uparrow$** | **CHAIR$_{\mathrm{S}}$ $\downarrow$** | **CHAIR$_{\mathrm{I}}$ $\downarrow$** | **SciQA $\uparrow$** |
> |---------------------|--------------------------------|--------------------------------|---------------------|--------------------------|---------------------------------------|---------------------------------------|---------------------|
> | **LLaVA-1.5**       | 1510.7                        | 348.2                         | 58.6                | 56.6                    | 54.3                                  | 11.3                                  | 66.8                |
> | **+ Vlfeedback**    | 1432.7                        | 321.8                         | 59.3                | 54.8                    | 40.3                                  | 13.2                                  | 66.2                |
> | **+ Human-Prefer**  | 1490.6                        | 335.0                         | 58.1                | 50.4                    | 38.7                                  | 11.3                                  | 65.8                |
> | **+ POVID**         | 1452.8                        | 325.3                         | 60.2                | 57.3                    | **35.2**                                  | **8.3**                               | **68.8**                |
> | **+ Fisao (Dino)**        | **1542.6**                    | **351.1**                     | **60.3**            | **61.5**                | 37.4                              | 9.3                                   | 68.7           |
>
>
> * b: Due to the time limitation we are unable to train Fisao using SigLip scores as rewards. However, to address the reviewer's concern, **we visualize the distributions of hallucinated and correct tokens using SigLip scores, as shown in Figure 12 in Appendix A.2.5. A detailed analysis has been included in  Appendix A.2.5 of the revised version.**   We observe that SigLip demonstrates better differentiation between hallucinated and correct tokens compared to CLIP. This suggests that our approach is potentially highly adaptable when applied to MLLMs incorporating multiple visual encoders.
> Regarding the modifications or considerations necessary to handle such configurations effectively, our framework can incorporate feedback from any visual encoder, including SigLip or others, by adhering to the reward calculation approach outlined in Equation (8). This adaptability makes our method versatile and effective across various settings and encoder architectures.

---

> ### Author Response · Authors · 2024-11-23
> **Response to Reviewer 3Rvt (4/4)**
>
> >Q6 Your approach primarily focuses on object hallucination. I am curious to know whether it also helps mitigate other types of hallucinations, such as those related to actions or spatial relationships. Additionally, could you elaborate on how the size of the expanded set C contributes to the observed performance improvements?
>
> **A6**    Thank you for your insightful question! We greatly appreciate your emphasis on exploring how our approach might mitigate various types of hallucinations, as this broadens the understanding of its potential impact.
> We will address your concerns one by one:
>
> **Your approach primarily focuses on object hallucination. I am curious to know whether it also helps mitigate other types of hallucinations, such as those related to actions or spatial relationships.**
>
> While our approach primarily targets object tokens, it inherently enhances the overall vision-language alignment in VLLMs, which can indirectly help reduce hallucinations related to actions or spatial relationships to some extent. By ensuring that the objects mentioned in the generated text are present in the image, the model develops a more accurate grounding in the scene, providing a foundation for better action and spatial relationship reasoning.
> We acknowledge that our method may not fully address hallucinations involving actions or relationships, as these areas rely on more complex contextual reasoning that pre-trained encoders are not specifically optimized for [1,2]. **We have addressed this limitation  in Appendix A.5 of the revised version and added it as an avenue for future exploration.**
>
>
> [1] Kuo W, Cui Y, Gu X, et al. F-vlm: Open-vocabulary object detection upon frozen vision and language models[J]. arXiv preprint arXiv:2209.15639, 2022.
>
> [2]  Lewis M, Nayak N V, Yu P, et al. Does clip bind concepts? probing compositionality in large image models[J]. arXiv preprint arXiv:2212.10537, 2022.
>
> **Additionally, could you elaborate on how the size of the expanded set C contributes to the observed performance improvements?**
>
> Regarding the expanded set $C$, we agree that its size plays a crucial role in the performance improvements. A larger and more comprehensive $C$ enables the model to more robustly identify and align object tokens, which enhances its ability to understand and generate contextually appropriate descriptions. However, when the set is expanded beyond a certain point, the performance improvements become limited.
> To substantiate this point, we compare the model's performance before and after the expansion of $C$ in Table R2. The results indicate that the expanded set $C$ leads to noticeable improvements in LVLM performance across various benchmarks.   However, the subsequent performance gains become marginal when we further add additional object labels to $C$. This suggests that the original set $C$ covers  a sufficiently comprehensive range of common objects, and further expansion may not significantly enhance the model’s capabilities.
>
> **Table R2: Performance Impact of Expanded Set $C$ on LVLM Benchmarks**
>
> | **Method**          | **MME^P** | **MME^C** | **SEED** | **MMB** | **MM-Vet** | **SQA** | **POPE** | **GQA** | **Cap_val** | **CHAIR$_S$** | **CHAIR$_I$** |
> |----------------------|---------------------|---------------------|----------|----------|-------------|---------------------|----------|----------|-------------|-----------------------|-----------------------|
> | Before expansion of $C$ |        1515.3           |        348.7             |     59.1     |   64.2    |    30.5    |              68.2     |        **85.9**    |     61.9      |     58.2      |       47.7                |           11.8            |
> | Original $C$   | 1522.6       | 349.0          | 60.6 | **64.8**     | **30.7**       | 69.3         | 85.7     | 62.0     | **61.2**    |**39.9**                 |**9.9**               |
> | After expansion of $C$  | **1536.1**        | **351.1**           | **60.9** | 64.7     | 30.6        | **69.8**            | 85.8     | **62.1**     | 61.0   | 41.1                  | 10.2                |
>
>
> These findings highlight that while expanding $C$ improves performance, the impact diminishes once the set becomes sufficiently inclusive of common objects.

---

> ### Author Response · Authors · 2024-11-25
> **Follow-up on Review Feedback**
>
> We sincerely thank you for your constructive comments, which have significantly improved the clarity and depth of our work. We have carefully addressed your concerns in the rebuttal, supplemented by additional detailed experimental results, visualizations, and expanded analyses, as summarized in Appendix A.2.2 and A.2.5. We look forward to your valuable feedback.

---

> ### Comment · Reviewer_3Rvt · 2024-11-26
> **Review decision.**
>
> My biggest concern is that the improvement due to the algorithm itself is limited. From the results in Table R2, the performance improvement appears marginal compared to the original LLaVA. A naive approach of augmenting the data used for preference fine-tuning with the Expanded Set C could potentially yield similar results. I am unable to identify a clear advantage over this naive approach. As a result, I will maintain the original score.

---

> > ### Author Response · Authors · 2024-12-02
> > **Response to Reviewer 3Rvt**
> >
> > Sorry for the delayed reply, and thank you for your thoughtful comments and feedback.
> > To address your concern:
> >
> > **"A naive approach of augmenting the data used for preference fine-tuning with the Expanded Set C could potentially yield similar results."**
> >
> > We adopt a naive approach by using only the average value of the ground truth (GT) as the threshold. Rewards are set to -1 for scores below the average value and 1 for scores above the average value.
> >
> > The experimental results are presented in Table R1 below.
> >
> > **Table R1: Performance Compared to the Naive Approach**
> >
> > | **Method**         | **MME^P** | **MME^C** | **SEED** | **MMB** | **MM-Vet** | **SQA** | **POPE** | **GQA** | **Cap_val** | CHAIR$_S$↓ | CHAIR$_I$↓ |
> > |--------------------|------------------|---------------------|----------|---------|------------|---------|----------|---------|------|------|--------------------|
> > | **FISAO**          | 1536.1       | 351.1          | 60.9 | 64.7    | 30.6       | 69.8| 85.8     | 62.1| 61.0 | 41.1 | 10.2|
> > | **Naive Approach** | 1513.5 | 340.9     | 55.4     | 60.3    | 29.8       | 60.5    | 83.7     | 58.3    | 58.2 | 48.6 | 12.5  |
> >
> > We can observe that FISAO outperforms the Naive Approach across various benchmarks. The Naive Approach uses a binary reward scheme (-1, 1), which oversimplifies the nuanced distribution of the vision encoder. In contrast, our method uses continuous reward signals avoiding these extremes. This allows the model to better  align with the vision encoder, resulting in improved performance and more precise feedback.

---

> > > ### Comment · Reviewer_3Rvt · 2024-12-03
> > > **Review decision.**
> > >
> > > Thank you for the detailed clarifications with experimental support. I would like to increase my review score.

---

> > > > ### Author Response · Authors · 2024-12-03
> > > > **Thanks!**
> > > >
> > > > Thank you for taking the time to reassess our work. We are deeply grateful for the opportunity to address your concerns and to improve our research based on your insightful feedback.

---

### Meta-Review · Area_Chair_SDcs · 2024-12-20

**Metareview:**

This paper proposes FiSAO (Fine-Grained Self-Alignment Optimization), a novel self-alignment method for vision-language models that uses token-level rewards to improve alignment without requiring additional data.

### Strengths:
1. Well-grounded theoretical framework
> "The methodological framework is well-defined, with both empirical and theoretical foundations" (Reviewer SJcY)
2. Novel conceptual contribution in token-level processing approach
> "analyzes the significant advantages of token-level processing over sentence-level processing in handling hallucinations, laying a solid foundation for subsequent research on token-level rewards" (Reviewer vDxF)

3. No additional data requirement and practical efficiency
> "The proposed self-training approach does not need any additional data and external modules to mitigate the misalignment issue" (Reviewer 3Rvt)

4. Comprehensive experimental validation
> "The experimental evaluation is thorough and well-designed. The experiments support the benefits of token-level rewards in reducing hallucinations and improving benchmark performance" (Reviewer ZUH4)

### Weaknesses:
1. Limited improvement on certain benchmarks
> "It is clear from Table 3 that your approach does not allow LLaVA-1.5 to surpass the higher-performing baselines on the MM-Vet benchmark" (Reviewer 3Rvt)

2. Initial theoretical presentation needed better justification
> "The theoretical analysis assumes that the CLIP provides perfect vision-langauge alignment, which is a bit unrealistic" (Reviewer ZUH4)

3. Computational overhead concerns
> "The paper contains no analysis of the performance - computational overhead tradeoffs against sentence-level rewards" (Reviewer SJcY)


### Justification:
While the paper has limitations on some benchmarks, it makes a valuable contribution to vision-language alignment research. The authors demonstrated strong responsiveness during rebuttal, addressing major concerns with new experiments and analyses. The method is practical (requiring no additional data) and shows competitive performance across most metrics. The computational efficiency analysis and expanded evaluations added during rebuttal strengthen the paper's practical value.

**Additional Comments On Reviewer Discussion:**

The authors provided detailed responses and substantial revisions:

1. Added new experiments with recent baselines (HA-DPO, RLAIF-V) showing competitive performance.

2. Clarified theoretical foundations with new proofs in Appendix A.3.2.

3. Addressed efficiency concerns with detailed benchmarks showing FiSAO achieves comparable or better computational efficiency versus POVID and VLFeedback when excluding inference time.

4. Improved evaluation using additional metrics like CIDEr and METEOR for better semantic comparison.

Notably, Reviewer 3Rvt and SJcY increased their assessment after these clarifications, while vDxF's concerns about data construction remained partially unaddressed due to lack of follow-up discussion.

---

### Decision · Program_Chairs · 2025-01-22

Accept (Poster)